# Mitigating Object Hallucination in Large Vision-Language Models through Adversarial Contrastive Finetuning

## Abstract

In recent years, large vision-language models (LVLMs) have made remarkable progress across a variety of vision-language tasks. However, they remain prone to object hallucination like generating descriptions of nonexistent objects in images. To explore the internal mechanism of object hallucination, we collected normal and hallucinated image-text pairs and performed quantitative analysis based on cosine similarity and qualitative analysis based on smooth Grad-CAM. We found that LVLMs may hallucinate due to incorrect extraction of image features and mismatch between image features and text features. Inspired by these findings, we propose an adversarial contrast fine-tuning (ACFT) method designed to enhance the alignment between visual and textual embedding and encourage the visual modality to focus on the correct image features, thus mitigating object hallucinations. The key approach involves automatically generating paired positive and negative examples using an adversarial hallucination attribute flipping (AHAF) method, followed by contrastive fine-tuning of the LVLM. Through extensive experiments, we show that ACFT achieves state-of-the-art performance on multiple benchmarks, e.g. outperforming existing approaches like VCD, OPERA and VTI, etc. on multiple benchmarks like POPE and MME.

## 1 Introduction

In recent years, Large Vision Language Models (LVLMs), such as LLaVA Liu et al. (2023), MiniGPT-4 Zhu et al. (2023), and GPT-4o OpenAI (2023), have achieved remarkable advancements. However, LVLMs still face significant challenges, particularly hallucination, which may lead to serious consequences in critical fields like medical diagnosis and autonomous driving. Effectively identifying and mitigating hallucinations in LVLMs has become an urgent research topic.In this study, we focus on a common hallucination type: object hallucination, where LVLMs either falsely "perceive" non-existent objects or "ignore" objects actually present in the image Rohrbach et al. (2018); Wei et al. (2024).

Given the complexity of LVLM systems, hallucinations may stem from multiple causes. Current explanations include over-reliance on language priors Leng et al. (2023); Zhu et al. (2024a); Chen et al. (2025), hallucination heads' dependence on text tokens Zhou et al. (2024); Yang et al. (2025a), tendency to generate new tokens by focusing on limited summary tokens Huang et al. (2024), and the absence of fine-grained reasoning supervision Zhang et al. (2024a), etc. The common view of these explanations is that they usually focus on exploring the causes of hallucinations from a text-prior perspective because they mainly focus on the long-output-text cases. However, we believe that the visual dimension also plays an important role in the generation of object hallucinations Sun et al. (2025), especially when the output text is short and the model's reasoning may rely more on the image dimension. Therefore, we try to focus on the image-prior perspective to explore the causes of hallucinations and corresponding mitigation methods in this study.

To explore the internal mechanism of object hallucination, we collected normal and hallucinated image-text pairs as positive and negative samples, respectively, and analyzed the differences between them both quantitatively and qualitatively. In quantitative analysis, we extract the image and text embeddings from a representative LVLM LLaVA v1.5 Liu et al. (2023), and compute the cosine

similarities between these embeddings for both positive and negative samples to reveal their representational differences within the LVLM embedding space. In qualitative analysis, we apply Smooth GradCAM Omeiza et al. (2019); Zhang et al. (2024b) techniques to visualize the attention maps of LVLMs on normal versus hallucinated images. The technical details can be found in *Supplementary Material (SM)*. The results are visualized, with a typical example illustrated in Figure 1. Through these analyses, we have two observations:

- The cosine similarity between the image and text embeddings of hallucinated samples is usually significantly lower than those of normal samples.
- The attention maps of hallucinated images tend to be dispersed outside the main object regions or concentrated in regions devoid of objects.

These preliminary findings indicate a potential misalignment between visual and textual modalities in current LVLMs, where visual modalities might fail to focus correctly on main objects or may incorrectly focus on non-existent objects. Therefore, the LVLM may cause hallucinations due to incorrect extraction of image features and mismatch between image and text features. We observe that this phenomenon usually occurs when textual prompts are relatively short, in which the model may tend to rely more on the visual modality.

Inspired by the above observations, we investigate contrastive learning to to enhance the alignment between visual and textual embeddings and encourage the visual modality to focus on the correct image features, thus mitigating hallucinations. A straightforward approach is ordinary contrastive fine-tuning (OCFT), in which we collect corresponding image-text pairs, treat the text as an anchor, the matched image as the positive sample, and a randomly selected unrelated image as the negative sample. The objective of OCFT is thus to minimize the cosine similarity between the anchor and the positive sample, while maximizing the similarity between the anchor and the negative sample.

However, we find that OCFT yields unsatisfactory results (see Experiments), likely due to the

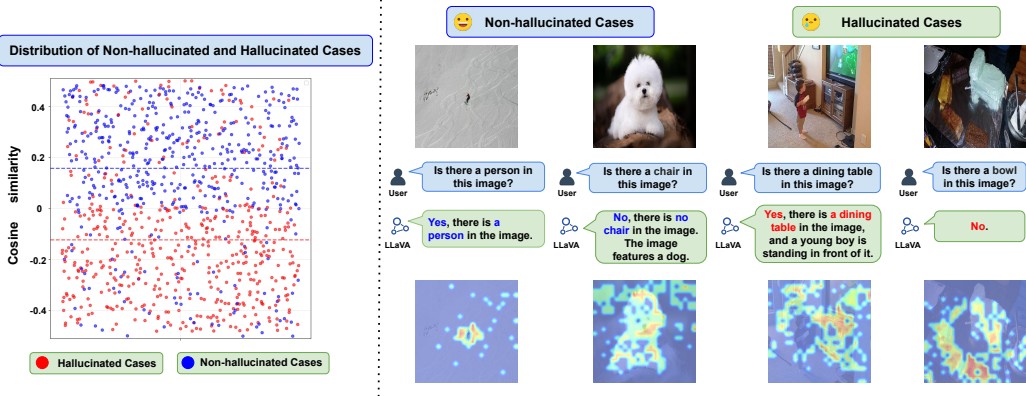

Figure 1: Analysis of object hallucination causes. **Left**: The cosine similarity distribution between the image and text embeddings of non-hallucinated (blue dots) and hallucinated (red dots) samples. Each dot in this image denotes the similarity between a pair of image and text embedding. The blue dashed line (- - -) shows the average cosine similarity of all non-hallucinated cases, while the red dashed line (- - -) shows the average similarity of hallucinated cases. **Right**: Case study using Smooth Grad-CAM. Blue annotations indicate responses without hallucinations, while red annotations highlight parts of the responses where hallucinations occur.

following limitation: the feature differences between positive and negative images in OCFT are highly uncontrolled and not focused specifically on the target object itself. Such lack of control makes it challenging for the model to learn consistent rules for attending to differences in the target object's features, thus impairing its ability to distinguish hallucination-inducing samples from non-hallucinating ones.

To address this challenge, we propose an adversarial contrast fine-tuning (ACFT) method. Our key innovation involves automatically generating aligned positive and negative examples using an adversarial hallucination attribute flipping (AHAF) approach, followed by contrastive fine-tuning of the LVLM. Compared to OCFT, ACFT offers clear advantages. First, in ACFT, each positive-negative image pair differs only by the controlled adversarial perturbation, the resulting samples

remain perfectly aligned, enabling precise, fine-grained analysis of the visual feature differences for the target object that give rise to hallucination. Second, ACFT injects adversarially optimized negative samples, which are tailored to exploit the target model's weaknesses, into the training process. This process helps to improve the model's ability to defend against strong perturbations and thereby learning more robust visual features that reduce hallucinations.

Compared to state-of-the-art hallucination mitigation methods, ACFT exhibits two advantages. First, compared to post-hoc correction methods applied during inference Yin et al. (2024); Wu et al. (2024), our method integrates directly into the training phase without increasing inference costs. Second, unlike full retraining methods Jiang et al. (2024) that utilize the entire dataset, our method requires only a small portion (approximately 0.9% of the entire COCO dataset Lin et al. (2015)), significantly reducing computational cost and training time.

Experimental results show that ACFT achieved better performance of mitigating object hallucination than various previous methods like VCD Leng et al. (2023), VTI Liu et al. (2024a), etc. on multiple benchmarks like POPE Li et al. (2023b) and MME Fu et al. (2024). Besides, ACFT did not compromise the model's original visual understanding performance and even slightly improved it in some cases.

## 2 RELATED WORK

### 2.1 LARGE VISION-LANGUAGE MODELS (LVLMS)

In recent years, Large Vision-Language Models (LVLMs) have advanced rapidly. Proprietary models such as OpenAI's GPT-4o OpenAI (2023) support both image and text inputs and show impressive performance across diverse multimodal tasks. Concurrently, numerous open-source LVLMs such as LLaVA Liu et al. (2023), MiniGPT-4 Zhu et al. (2023), etc. have been introduced. Most of these models adopt a "visual encoder + large language model" architecture, achieving cross-modal alignment and reasoning capabilities through pretraining and instruction tuning.

### 2.2 MITIGATING HALLUCINATIONS FOR LVLMS

Hallucination, where models generate descriptions inconsistent with input images, remains a major challenge for LVLMs. Various strategies have been proposed to mitigate this issue including input-level-decoding Leng et al. (2023); Huang et al. (2024), post-processing Yin et al. (2024), latent-space-processing Liu et al. (2024a) methods, etc. Most of them are language-prior methods and focus on the long-output-text cases. For example, VCD Leng et al. (2023) suppress hallucination-prone tokens by comparing outputs from original versus perturbed images or biased decoding branches. OPERA Huang et al. (2024) introduces overconfidence penalties and rollback mechanisms during decoding to reduce reliance on linguistic priors. However, although these language-prior methods perform well in long-output-text settings, their effectiveness is unsatisfactory in short-output-text settings, which motivates us to explore hallucination mitigation from an image-prior perspective, especially for short-output-text scenarios.

## 3 METHODS

### 3.1 MITIGATING OBJECT HALLUCINATION

To mitigate object hallucination of LVLMs, we propose a two-stage framework as shown in Figure 2. Stage one is AHAF, where we apply subtle adversarial perturbations on original images to construct aligned positive-negative image pairs. Stage two is ACFT, in which we design an adversarial contrastive loss function and fine-tune the LVLMs to mitigate object hallucination.

### 3.2 ADVERSARIAL HALLUCINATION ATTRIBUTE FLIPPING

To facilitate effective contrastive learning, we need to construct aligned positive–negative image pairs. In this study, we define a positive sample as an image that does not induce hallucinations in the model's output, and a negative sample as one that does. While one could manually collect natural

images as positive and negative examples for contrast learning, doing so yields unsatisfactory results (See Section *Comparision with OCFT*). We believe the reason is that the feature differences between positive and negative images in OCFT are highly uncontrolled and not focused specifically on the target object itself, which makes it challenging for the LVLM to for attending to differences in the target object's features, thus impairing its ability to distinguish hallucination-inducing samples from non-hallucinating ones. To address this challenge, we propose an AHAF method to automatically construct aligned positive-negative image pairs and selectively alter key visual features for triggering targeted object hallucinations.

As illustrated in Figure 2, AHAF first applies subtle adversarial perturbations generated by PGD Madry et al. (2019) method to the original images with its adversarial loss (Equation 3). These perturbations are then optimized to flip their hallucination attributes, such as converting a non-hallucinating image into one that induces object hallucination, and vice versa. The AHAF method thus automatically generates aligned positive–negative sample pairs in a highly targeted and efficient way. Because each pair differs only by the controlled adversarial perturbation, the resulting samples remain perfectly aligned, enabling precise, fine-grained analysis of the visual feature differences for the target object that gives rise to hallucination. Next, we provide a detailed description of

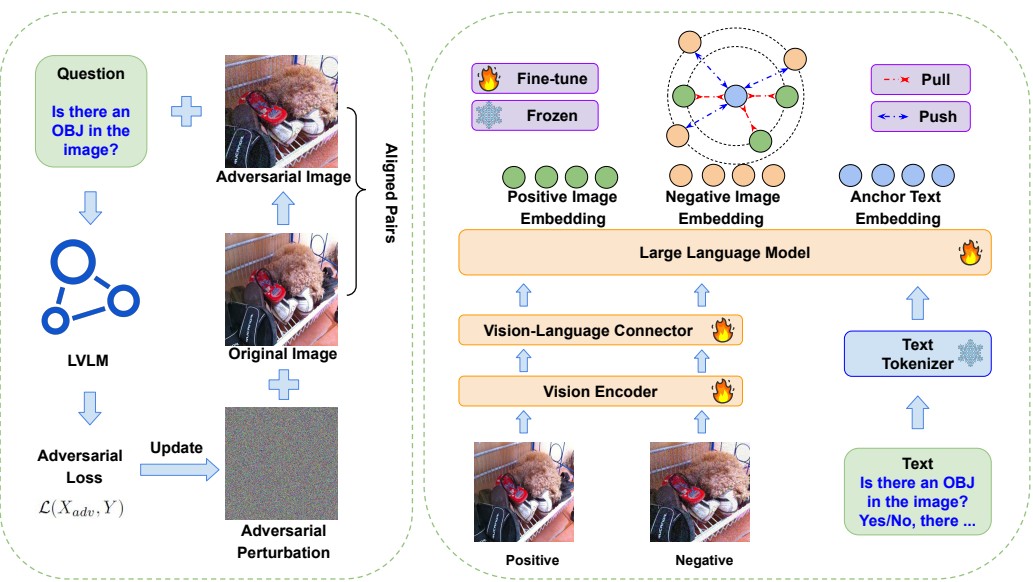

Figure 2: Pipeline of our method. **Left**: Pipeline of AHAF. We apply subtle adversarial perturbations on original images to construct aligned positive-negative image pairs. **Right**: Pipeline of ACFT. The core of ACFT is to maximize the similarity between the text anchor and the positive image sample, while minimizing the similarity between the text anchor and the negative image sample.

the AHAF method. Given an original image $X$ and a target text set $Y = \{y_i\}_{i=1}^m$, our objective is to generate an adversarial image $X_{adv}$ that, when input to the model, maximizes the model's likelihood of giving the target output. The optimization objective is formulated as:

$$X_{adv} := \arg\min_{\hat{X}_{adv} \in \mathcal{B}} \sum_{i=1}^m -\log\left(p(y_i|\hat{X}_{adv})\right), \tag{1}$$

To find suitable perturbations within the constrained space $\mathcal{B}$, PGD updates the image through the following iterative process:

$$X_{adv}^{t+1} = \Pi_{\mathcal{B}(X,\epsilon)}\left(X_{adv}^t + \alpha \cdot \text{sign}\left(\nabla_{X_{adv}^t} \sum_{i=1}^m -\log\left(p(y_i|X_{adv}^t)\right)\right)\right), \tag{2}$$

In practice, we employ the cross-entropy loss to quantify the divergence between the model's output and the target text:

$$\mathcal{L}(X_{adv}, Y) = -\sum_{i=1}^m \log\left(p(y_i|X_{adv})\right), \tag{3}$$

After multiple iterations, the image $X_{adv}$ is used as a contrastive sample that can induce the model to produce an answer opposite to that of the original image.

### 3.3 ADVERSARIAL CONTRASTIVE FINE-TUNING

Inspired by our preliminary findings (in Figure 1), we propose an ACFT method designed to enhance the alignment between visual and textual embedding and encourage the visual modality to focus on the correct image features, thus mitigating hallucinations. Based on the aligned positive–negative image pairs generated using AHAF method, the core of ACFT is to maximize the similarity between the text anchor and the positive image sample, while minimizing the similarity between the text anchor and the negative image sample in the embedding space through self-supervised contrast learning. The process of the ACFT algorithm is outlined in Algorithm 1.

We design the ACFT method based on the following considerations. First, inspired by adversarial training Madry et al. (2018); Bai et al. (2021), we inject adversarially optimized negative samples, which are tailored to exploit the target model's weaknesses, into the training process. The goal is to improve the model's ability to defend against strong perturbations and thereby learning more robust visual features that reduce hallucinations. Many studies Li et al. (2024; 2023a); Liu et al. (2025) have shown that adversarial training enhances model robustness to not only adversarial noise but also natural perturbations (e.g. illumination changes, blur, etc.). Second, drawing on contrastive learning Radford et al. (2021); Jiang et al. (2024) principles, we construct aligned positive–negative sample pairs and apply a self supervised contrastive objective to sharpen the model's discrimination between hallucination inducing and non hallucinating images. Third, contrastive fine tuning offers strong generality, making ACFT applicable to a wide range of backbone architectures and ensuring both transferability and scalability. Finally, unlike full dataset retraining, our fine tuning approach requires only a modest amount of data, and unlike post post-processing technique, it imposes no extra inference-time overhead.

According to the above principles, we design an adversarial contrastive loss function $L_{\text{contra}}$ to measure similarity differences between positive and negative samples. To compute $L_{\text{contra}}$, we define a similarity-based contrastive objective between text and image representations. Specifically, for each anchor text $T_i$, we construct a corresponding positive-negative image pair $(X_i^+, X_i^-)$, where $X_i^+$ aligns with the text, while $X_i^-$ induces hallucination. Using the visual encoder $f(X)$ and text encoder $g(T)$, we extract image and text embeddings as $z^+ = f(X^+)$, $z^- = f(X^-)$ and $t = g(T)$, respectively. The adversarial contrastive loss for a single training instance is defined as:

$$\ell_{\text{contra}}^{(i)} = -\log(\exp\left(\psi(t_i, z_i^+)/\tau\right) + \log(\exp\left(\psi(t_i, z_i^+)/\tau\right) + \exp\left(\psi(t_i, z_i^-)/\tau\right)), \quad (4)$$

where $\tau$ is a hyperparameter that controls the sharpness of the softmax distribution. The similarity between two vectors is defined as:

$$\psi(\mathbf{x}, \mathbf{y}) = \frac{\mathbf{x}^\top \mathbf{y}}{|\mathbf{x}| \cdot |\mathbf{y}|}. \quad (5)$$

In batch training, where each batch contains $N$ text-image pairs, the batch-wise adversarial contrastive loss $L_{\text{contra}}$ is:

$$L_{\text{contra}} = \frac{1}{N} \sum_{i=1}^{N} \ell_{\text{contra}}^{(i)}, \quad (6)$$

While mitigating hallucinations, we also aim to preserve the LVLM's original visual-language generation capability. To this end, we introduce a classical cross-entropy generation loss $L_{\text{gen}}$ during fine-tuning. This loss measures the divergence between the model's predicted token distribution and the true distribution. Specifically, for a target sequence of length $T$, the model predicts the probability of the ground-truth token $y_t^*$ conditioned on the preceding outputs $y_{<t}$ and the image features $I$ at each time step $t$. We then compute

$$L_{\text{gen}} = -\frac{1}{T} \sum_{t=1}^{T} \log p_\theta\left(y_t^* \mid y_{<t}, X\right), \quad (7)$$

which averages the negative log-likelihood of the correct tokens over the entire sequence.

The overall objective is defined as:

$$L_{\text{total}} \ = \ L_{\text{gen}} \ + \ \lambda \, L_{\text{contra}} \,, \tag{8}$$

where $\lambda$ is a hyperparameter that is determined empirically.

---

**Algorithm 1** Adversarial Contrast Fine-Tuning

---

**Input**: A batch of text inputs $\{T_i\}_{i=1}^{N}$ and corresponding positive–negative image pairs $\{(X_i^+, X_i^-)\}_{i=1}^{N}$
**Parameter**: Temperature $\tau$, contrastive loss weight $\lambda$, visual encoder $f(\cdot)$, text encoder $g(\cdot)$
**Output**: Fine-tuned model parameters

1: **for** each training iteration **do**
2:    **for** each triplet $(T_i, X_i^+, X_i^-)$ in batch **do**
3:       Extract embeddings: $t_i = g(T_i)$, $z_i^+ = f(X_i^+)$, $z_i^- = f(X_i^-)$.
4:       Compute similarities: $\text{sim}(t_i, z_i^+)$ and $\text{sim}(t_i, z_i^-)$.
5:       Compute contrastive loss $\ell_{\text{contra}}^{(i)}$ using Equation equation 4.
6:    **end for**
7:    Compute batch contrastive loss using Equation equation 6.
8:    **for** each $(T_i, X_i^+)$ in batch **do**
9:       Perform generation task to compute $L_{\text{gen}}$ via Equation equation 7.
10:    **end for**
11:    Combine losses: $L_{\text{total}} = L_{\text{gen}} + \lambda L_{\text{contra}}$.
12:    Backpropagate and update model parameters using $L_{\text{total}}$.
13: **end for**
14: **return** Fine-tuned model.

---

## 4 EXPERIMENTS

### 4.1 TARGET LVLMS

To verify the effectiveness of our method, we adopted two representative LVLMs: LLaVA v1.5-7B Liu et al. (2023) and MiniGPT-4 13B Zhu et al. (2023) as the target LVLMs. Both models have strong end-to-end vision-language understanding and generation capabilities, and are widely used as target models in previous research Leng et al. (2023); Huang et al. (2024); Yin et al. (2024); Liu et al. (2024b).

### 4.2 BASELINE METHODS

We compared our ACFT method with four state-of-the-art baseline methods, including two typical input-level decoding methods: Visual Contrastive Decoding (VCD) Leng et al. (2023) and OPERA Huang et al. (2024), one typical post-processing method: Woodpecker Yin et al. (2024), and one typical latent-space-processing method: Visual and Textual Intervention (VTI) Liu et al. (2024a).

### 4.3 BENCHMARKS

We evaluated all methods using two benchmarks: POPE Li et al. (2023b) and MME Fu et al. (2024).

**POPE** is a benchmark specifically designed to assess object hallucinations in images. It formulates hallucination assessment as a binary classification task: given an image and an object, the model is asked simple queries "Is there an OBJ in the image?". POPE includes three subsets based on object sampling strategies: (1) **Random**: randomly selected objects from the COCO dataset Lin et al. (2015); (2) **Popular**: objects frequently appearing in training data or common scenes; (3) **Adversarial**: objects highly related to those present in the image but actually absent. We use the official benchmark of POPE, which includes 3,000 question-answer pairs for each subset.

**MME** is a comprehensive benchmark for evaluating LVLMs across 14 tasks spanning perception and cognition. Among them, the **Existence** subset is most relevant to object hallucination: it requires models to judge whether a given object or attribute exists in the image, typically answering with "Yes" or "No"— same as POPE. Beyond the existence subset, MME also covers multiple tasks such as counting, localization, OCR, and commonsense reasoning.

| Subset | Method | LlaVA v1.5 7B | | | | MiniGPT4 13B | | | |
|--------|--------|------|------|------|------|------|------|------|------|
| | | *ACC* | *Pre.* | *Rec.* | *F1* | *ACC* | *Pre.* | *Rec.* | *F1* |
| *Adversarial* | origin | 0.779 | 0.721 | 0.911 | 0.805 | 0.700 | 0.670 | 0.791 | 0.725 |
| | VCD | 0.808 | **0.847** | 0.753 | 0.797 | 0.734 | 0.701 | 0.817 | 0.754 |
| | OPERA | 0.798 | 0.787 | 0.816 | 0.802 | 0.737 | 0.736 | 0.738 | 0.737 |
| | Woodpecker | 0.771 | 0.710 | **0.917** | 0.800 | 0.741 | 0.678 | **0.917** | **0.780** |
| | VTI | 0.805 | 0.770 | 0.871 | 0.817 | 0.700 | 0.668 | 0.795 | 0.726 |
| | Ours | **0.841** | 0.802 | 0.905 | **0.850** | **0.771** | **0.811** | 0.708 | 0.756 |
| *Popular* | origin | 0.862 | 0.832 | 0.905 | 0.867 | 0.732 | 0.709 | 0.787 | 0.747 |
| | VCD | 0.882 | **0.917** | 0.839 | 0.876 | 0.748 | 0.746 | 0.752 | 0.749 |
| | OPERA | 0.886 | 0.847 | **0.940** | 0.891 | 0.737 | 0.715 | 0.789 | 0.750 |
| | Woodpecker | 0.789 | 0.734 | 0.906 | 0.811 | 0.765 | 0.706 | **0.908** | 0.794 |
| | VTI | 0.868 | 0.842 | 0.908 | 0.874 | 0.722 | 0.691 | 0.804 | 0.743 |
| | Ours | **0.906** | 0.907 | 0.905 | **0.906** | **0.818** | **0.910** | 0.707 | **0.795** |
| *Random* | origin | 0.885 | 0.867 | 0.910 | 0.888 | 0.792 | 0.792 | 0.792 | 0.792 |
| | VCD | 0.892 | 0.881 | 0.906 | 0.893 | 0.808 | 0.769 | 0.881 | 0.821 |
| | OPERA | 0.878 | **0.918** | 0.831 | 0.873 | 0.817 | 0.819 | 0.814 | 0.816 |
| | Woodpecker | 0.834 | 0.788 | **0.914** | 0.846 | 0.818 | 0.766 | **0.917** | **0.835** |
| | VTI | 0.891 | 0.906 | 0.870 | 0.888 | 0.799 | 0.761 | 0.871 | 0.812 |
| | Ours | **0.897** | 0.890 | 0.905 | **0.897** | **0.843** | **0.972** | 0.705 | 0.818 |

Table 1: Results comparison on the POPE benchmark.

We selected these two benchmarks because all their questions are answered with either 'Yes' or 'No'. We believe the model's reasoning may rely more on image dimension when the output text is relatively short. This short-text-output setting aligns well with our focus on the image-prior perspective to explore the causes of hallucinations and corresponding mitigation methods in this study. In contrast, many previous research Leng et al. (2023); Zhu et al. (2024b) focused on the long-text-output settings (like "describe this image in detail") that align with their language-prior perspective to explore the causes of hallucinations.

### 4.4 EVALUATION METRICS

In our experimental setup, all questions were answered with either 'Yes' or 'No'. Therefore, whether the model hallucinated can be formulated as a classification problem. We chose four widely used classification metrics including *accuracy*, *precision*, *recall*, and *F1 score* for evaluation.

### 4.5 IMPLEMENTATION DETAILS

We present the implementation details of ACFT and baseline methods like adversarial attack settings, finetuning strategies, hyperparameter settings, GPU, etc. in Appendix A.2 and A.3.

### 4.6 COMPARISION BETWEEN OCFT AND ACFT

In this section, we compare the performance of OCFT and ACFT in mitigating object hallucination of LVLMs. We use the same 3,000 images from the COCO dataset Lin et al. (2015) as the training set. For ACFT, we employ AHAF to generate aligned positive-negative image pairs for contrast finetuning. In contrast, OCFT constructs image pairs by randomly sampling a "positive" image that matches a given text anchor (e.g., "cat") and a "negative" image drawn arbitrarily from

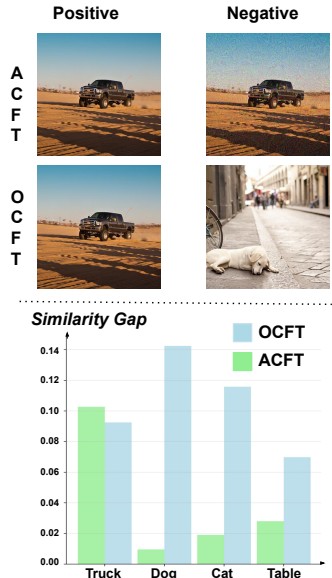

Figure 3: Comparison between OCFT and ACFT. **Top**: One Example of positive and negative samples for ACFT and OCFT. The target object is "truck". **Bottom**: Cosine similarity gaps between positive and negative samples for the target object and non-target objects.

other categories (e.g. "dog"), resulting in unaligned and semantically inconsistent pairs. Both methods use the same fine-tuning strategy on LLaVA v1.5 7B, and are evaluated on three subsets of the POPE benchmark.

As shown in Table 3, ACFT significantly outperformed OCFT in mitigating object hallucination, with accuracy improvements of 35.8%, 7.4%, and 17.6% on the three subsets, respectively. Notably, OCFT performed particularly poorly on the *Adversarial* subset, only achieving an accuracy of 0.483, which was even much lower than that of the original LLaVA model. These results highlight the effectiveness of ACFT compared to OCFT.

We computed the similarity gaps between positive and negative samples for the target object and non-target objects, respectively, using Equation equation 9. Assuming the embeddings of the positive image, negative image, and anchor text are $z^+$, $z^-$, and $t$, respectively, we define the similarity gap $\Delta$ as:

$$\Delta(z^+, z^-, t) = |\psi(z^+, t) - \psi(z^-, t)|, \tag{9}$$

where the cosine similarity $\psi$ between two vectors is defined in Equation 5. A representative example is shown in Figure 3. More experimental details and results are provided in Appendix A.4. These results indicate that for ACFT, the similarity gap between positive and negative samples for the target object is clearly distinguishable from that for non-target objects. This property facilitates the model in learning consistent rules to focus on differences in the target object's features, thereby enhancing its ability to distinguish hallucination-inducing samples from non-hallucinating ones. However, OCFT lacks this property, which partially explains why ACFT achieves superior performance compared to OCFT.

| Model | Method | ACC | Precision | Recall | F1 Score |
|-------|--------|-----|-----------|--------|----------|
| LlaVA v1.5 | origin | 0.950 | 0.909 | **1.000** | 0.952 |
| | VCD | 0.950 | 0.935 | 0.967 | 0.951 |
| | OPERA | 0.933 | 0.964 | 0.900 | 0.931 |
| | Woodpecker | 0.933 | 0.882 | 1.000 | 0.937 |
| | VTI | 0.967 | 0.967 | 0.967 | 0.967 |
| | Ours | **0.983** | **1.000** | 0.967 | **0.983** |
| MiniGPT4 | origin | 0.850 | 0.800 | **0.933** | 0.861 |
| | VCD | 0.867 | 0.867 | 0.867 | 0.867 |
| | OPERA | 0.850 | 0.838 | 0.867 | 0.852 |
| | Woodpecker | 0.833 | 0.794 | 0.900 | 0.843 |
| | VTI | 0.883 | 0.896 | 0.867 | 0.881 |
| | Ours | **0.900** | **0.900** | 0.900 | **0.900** |

Table 2: Results comparison on the MME Existence subset.

| Subset | Method | ACC | Precision | Recall | F1 Score |
|--------|--------|-----|-----------|--------|----------|
| *Adversarial* | OCFT | 0.483 | 0.489 | 0.784 | 0.602 |
| | ACFT | **0.841** | **0.802** | **0.905** | **0.850** |
| *Popular* | OCFT | 0.832 | 0.867 | 0.784 | 0.824 |
| | ACFT | **0.906** | **0.907** | **0.905** | **0.906** |
| *Random* | OCFT | 0.721 | 0.696 | 0.784 | 0.737 |
| | ACFT | **0.897** | **0.890** | **0.905** | **0.897** |

Table 3: Comparison between OCFT and ACFT.

| Model | Method | ACC | Precision | Recall | F1 Score |
|-------|--------|-----|-----------|--------|----------|
| LlaVA v1.5 | Origin | 0.728 | 0.666 | 0.916 | 0.771 |
| | Ours | **0.747** | **0.683** | **0.922** | **0.785** |
| MiniGPT4 | Origin | 0.538 | 0.531 | 0.655 | 0.586 |
| | Ours | **0.548** | **0.538** | **0.672** | **0.598** |

Table 4: Results comparison on the MME whole benchmark.

### 4.7 EFFECTIVENESS OF ACFT ON MITIGATING HALLUCINATION

We compared ACFT with other baseline methods on POPE and MME-Existence benchmark to show its effectiveness on mitigating hallucination.

#### 4.7.1 EVALUATION ON POPE

Table 1 shows that ACFT significantly outperforms all baseline methods across two different target LVLMs. For the LlaVA model, ACFT achieved accuracies of 0.841, 0.906, and 0.897 on the three subsets of POPE, surpassing the second-best baseline by 3.3%, 2.0%, and 0.5%, respectively. Similarly, for the MiniGPT-4 model, ACFT achieved the best accuracy, surpassing the second-best baseline by 3.0%, 5.3%, and 2.5%, respectively.

The advantage stems from ACFT's adversarial contrastive pairs during training, which enhance the alignment between visual and textual embedding and encourage the visual modality to focus on the correct image features, thus mitigating hallucinations. In contrast, VCD encourages the model to focus on the text output, which limits its performance of mitigating image-induced hallucinations. Similarly, OPERA is also a language-prior method that penalizes overconfident decoding paths when the model focuses on summary tokens. VTI adjusts latent features only at inference and lacks sufficient generalization performance. Woodpecker relies on external grounding modules that may misdetect objects. These limitations constrain baseline performance on vision-dependent tasks, whereas ACFT's focused visual alignment yields a clear performance improvement.

### 4.7.2 EVALUATION ON MME-EXISTENCE

The results in Table 2 showed that ACFT achieved the best performance on the MME-Existence sub-set. For the LLaVA model, although the original version already attained a relatively high accuracy of 0.950, ACFT further improved it by 3.3%. In contrast, other baseline methods such as OPERA and Woodpecker, even degraded the model's performance. For the MiniGPT4 model, ACFT brought an improvement, 1.7%, compared to the best-performing baseline. These results highlight the robustness and effectiveness of ACFT across different target models and benchmarks.

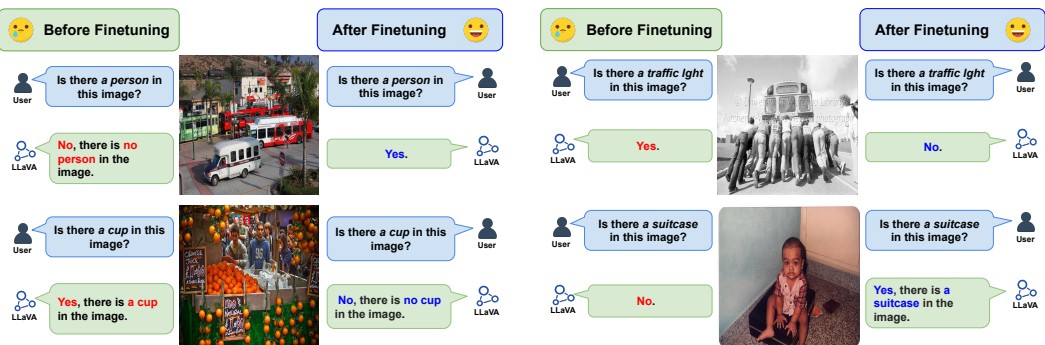

Figure 4: Visualization results of our proposed method. Blue annotations indicate responses without hallucinations, while red annotations highlight parts of the responses where hallucinations occur.

### 4.8 EFFECTIVENESS OF ACFT ON VISUAL COMPREHENSION

A Previous study Liu et al. (2024d) showed that fine-tuning on specific tasks may compromise a model's general capabilities. Thus, it remains a concern whether our method, ACFT, might impair the model's general visual comprehension ability, even if it effectively mitigates hallucination. In this section, we evaluated ACFT on the full MME benchmark, which comprehensively assesses a model's visual comprehension. As shown in Table 4, ACFT did not impair performance; instead, it slightly improved the target VLMs' overall score. These results indicate that ACFT not only mitigates hallucinations but also enhances the model's robustness in visual feature extraction and comprehension, thereby improving its general visual understanding ability.

### 4.9 ABLATION STUDY

To verify the effectiveness of our ACFT loss $L_{\mathrm{contra}}$, we conducted an ablation study, as detailed in Appendix A.5. The results highlight the effectiveness of the proposed adversarial contrastive loss $L_{\mathrm{contra}}$ to mitigate object hallucination, particularly under challenging or misleading conditions.

### 4.10 VISUALIZATION

We present visualized examples of ACFT. As shown in Figure 4, after applying ACFT, LVLMs no longer hallucinate. The improvement is evident when the target object occupies a small region of the image, suggesting that ACFT enhances the model's ability to capture fine-grained visual features.

Further analysis (detailed in Appendix A.6) shows that, for samples that induced hallucinations, ACFT improved the cosine similarity between their image and text embeddings. And the models' Grad-CAM attention maps became more tightly focused on the target objects. This, to some extent, explains the underlying mechanism of ACFT's effectiveness and confirms our initial analysis.

## 5 CONCLUSION

This paper addresses the problem of object hallucination in large vision-language models (LVLMs). Through both quantitative and qualitive analysis, we found that object hallucinations in LVLMs stem from incorrect extraction of image features and mismatch between image features and text features. Inspired by these findings, we propose an ACFT method to mitigate object hallucination. The key approach involves automatically generating aligned positive and negative examples using an AHAF method, followed by contrastive fine-tuning of the LVLMs. Experimental results show that ACFT achieves state-of-the-art performance on multiple benchmarks like POPE and MME.

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

# A APPENDIX

## A.1 DETAILED ANALYSIS OF HALLUCINATION CAUSES

We analyze the underlying causes of object hallucination in LVLMs from the perspective of visual features. We use LLaVA v1.5 7B as a representative LVLM in the following analysis.

### A.1.1 QUANTITATIVE ANALYSIS

To quantitatively assess the alignment between visual feature and ground truth text, we compute the cosine similarity between the image embedding $\mathbf{v}$ and the corresponding ground truth text embedding $\mathbf{t}$ of LVLM. The cosine similarity is calculated as:

$$\text{CosineSim}(\mathbf{v}, \mathbf{t}) = \frac{\mathbf{v} \cdot \mathbf{t}}{|\mathbf{v}| \cdot |\mathbf{t}|}, \tag{10}$$

Higher similarity indicates better alignment between visual features and textual semantics. By comparing the average similarity across correctly predicted and hallucinated cases, we evaluate how the alignment between visual features and text semantics correlates with hallucination occurrences.

In the analysis, we do not measure similarity between the image and an arbitrary input question. Instead, for each image, we first construct 10 POPE-style questions(Is there a [object] in the image?) based on the ground-truth annotations in COCO, and prompt the model to answer them. If all questions are answered correctly, we label the image as non-hallucinated; otherwise, we label it as hallucinated. Next, we randomly sample 500 correct cases and 500 hallucinated cases from COCO dataset Lin et al. (2015). For each sampled image, we take its ground-truth caption and compute the cosine similarity between the image embedding and ground-truth text embedding. As shown on the left of Figure 1, correct cases exhibit significantly higher text-image embedding similarity compared to hallucinated cases. The average similarity for correct cases is 0.158, while for hallucinated cases it drops to -0.122. This clear gap suggests that when hallucination occurs, the model's image representation is less aligned with the ground truth text, indicating inaccurate visual perception.

### A.1.2 QUALITATIVE ANALYSIS

In qualitative analysis, we adopt Smooth Grad-CAM Zhang et al. (2024b) techniques to visualize the attention maps of LVLMs on non-hallucinated versus hallucinated images.

Given the output logits $\mathbf{z} = [z_1, z_2, \ldots, z_n]$, we sum the logits to obtain:

$$z_{\text{answer}} = \sum_{i=1}^{n} z_i \tag{11}$$

For the target layer's feature maps $A^k$, we compute the gradients:

$$G^k = \frac{\partial z_{\text{answer}}}{\partial A^k} \tag{12}$$

Then, global average pooling derives channel weights:

$$\alpha_k = \frac{1}{Z} \sum_{i,j} G^k_{i,j} \tag{13}$$

where $Z$ is the spatial resolution of $A^k$.

The Grad-CAM map is calculated as:

$$M_{\text{Grad-CAM}} = \text{ReLU}\left(\sum_k \alpha_k A^k\right) \tag{14}$$

For Smooth Grad-CAM, input image noise $\epsilon_i \sim \mathcal{N}(0, \sigma^2)$ is added, and the Grad-CAM maps are averaged over $N$ samples:

$$M_{\text{Smooth Grad-CAM}} = \frac{1}{N} \sum_{i=1}^{N} M^{(i)}_{\text{Grad-CAM}} \tag{15}$$

We further analyze the model's internal attention using Smooth Grad-CAM. The right side of Figure 1 presents attention heatmaps for two non-hallucinated and two hallucinated examples. In non-hallucinated cases, the model focuses accurately on the main object, so that it correctly recognizes existing objects and points out nonexistent objects. In contrast, hallucinated cases show two distinct failure patterns: (1) the model incorrectly focuses on objects visually similar to the target and perceives nonexistent objects (third case); (2) the model focuses on irrelevant regions, neglecting the actual object (fourth case). These patterns reveal that object hallucination often stems from misdirected or insufficient attention to relevant visual features.

### A.1.3 SUMMARY

Both quantitative and qualitative results show that object hallucination in LVLMs is closely linked to incorrect visual feature perception. Compared to correct cases, hallucinated instances often have lower text-image embedding similarity and misaligned attention distributions. Inspired by these findings, we propose an adversarial fine-tuning framework that explicitly contrasts hallucinated and correct cases to help the model learn more accurate visual representations and mitigate object hallucination.

### A.2 IMPLEMENTATION DETAILS FOR BASELINE METHODS

In this section, we also provide the detailed hyperparameter settings used for each baseline. Most hyperparameters are kept consistent with those reported in the original papers. For VCD, we set the noise step to 500, `cd-alpha` to 1, and `cd-beta` to 0.1. For OPERA, we use a scale factor of 50.0, set the OPERA threshold to 15, the number of attention candidates to 5, the penalty weights to 1.0, and apply beam search with 5 beams. For Woodpecker, we adopt GroundingDINO Liu et al. (2024c) as the detector model, setting the box threshold to 0.35 and the text threshold to 0.25. For VTI, we set $\alpha$ to 0.2, $\beta$ to 0.4, the beam search size to 4, and the mask ratio to 0.99.

### A.3 IMPLEMENTATION DETAILS FOR AHAF AND ACFT

We present the detailed hyperparameters used in AHAF and ACFT. For AHAF, we mainly employed PGD to generate adversarial perturbation and set the number of iterations to 100, the attack budget $\epsilon$ to 16 / 255, and the step size $\alpha$ to 1. For ACFT, we fine-tuned the target LVLMs using LoRA Hu et al. (2021). We kept the text tokenizer frozen and fine-tuned the language model, vision-language connector, and vision encoder. We trained LVLMs on 3,000 samples for 2 epochs with a batch size of 16. The learning rates are set to 2e-4 for the language model, 2e-5 for the vision-language connector, and 1e-5 for the vision encoder. For $\lambda$ in Equation(8), we conduct a hyperparameter search in the range 0.1, 0.2, 0.25, 0.3, 0.4, 0.5, 1.0, and set $\lambda = 0.25$ based on the empirical results. For $\tau$ in Equation(4), we also conduct a hyperparameter search in the range 0.01, 0.03, 0.05, 0.07, 0.1, 0.2, and set $\tau = 0.05$ based on the empirical results. All training is conducted on a single NVIDIA H100 GPU.

### A.4 EXPERIMENTAL SETUP AND RESULTS FOR COMPARISION BETWEEN OCFT AND ACFT

In this section, we present the experimental setup and supplementary results for Section: *Comparision between OCFT and ACFT*.

We randomly sampled 50 images from the COCO dataset Lin et al. (2015) as positive images for both methods. For ACFT, we applied PGD attack to generate corresponding negative samples. For OCFT, negative samples were randomly selected from the COCO dataset.

We used LLaVA v1.5 Liu et al. (2023) to extract embeddings for both positive and negative samples, denoted as $z_A^+$, $z_A^-$, $z_O^+$, and $z_O^-$, respectively. Then we selected a target object (i.e., the object intended to induce hallucination) along with several irrelevant non-target objects. For each, we obtained the corresponding anchor text embeddings, denoted as $t^{tar}$ and $t^{non}$. We then computed the cosine similarity between each image embedding and text embedding using cosine similarity $\psi(t, z)$, and get $sim^+ = \psi(t, z^+)$ and $sim^- = \psi(t, z^-)$. The similarity gap $\Delta$ was calculated as $\Delta = |sim^+ - sim^-|$.

| Subset | Loss | ACC | Precision | Recall | F1 Score |
|--------|------|-----|-----------|--------|----------|
| *Adversarial* | w/o $L_{\text{contra}}$ | 0.797 | 0.743 | **0.906** | 0.817 |
| | w/ $L_{\text{contra}}$ | **0.841** | **0.802** | 0.905 | **0.850** |
| *Popular* | w/o $L_{\text{contra}}$ | 0.862 | 0.832 | **0.906** | 0.867 |
| | w/ $L_{\text{contra}}$ | **0.906** | **0.907** | 0.905 | **0.906** |
| *Random* | w/o $L_{\text{contra}}$ | 0.896 | 0.888 | **0.906** | 0.897 |
| | w/ $L_{\text{contra}}$ | **0.897** | **0.890** | 0.905 | **0.897** |

Table 5: Ablation study of ACFT loss.

For ACFT, the average similarity gap for target object $\Delta_A^{tar}$ is 0.738, while the average similarity gap for non-target objects $\Delta_A^{non}$ is 0.276, which shows an evident distinction. And the similarity gap of target objects is clearly much higher than non-target ones. In contrast, OCFT yields $\Delta_O^{tar} = 0.104$ and $\Delta_O^{non} = 0.121$, which are nearly indistinguishable. These results suggest that ACFT can effectively manipulate the model's perception of the target object while exerting minimal influence on non-target objects—an ability that OCFT fails to achieve. The results partially explain why ACFT achieves superior performance compared to OCFT.

## A.5 ABLATION STUDY DETAILS

To verify the effectiveness of our ACFT loss $L_{\text{contra}}$, we conducted an ablation study. We fine-tuned LLaVA v1.5 7B using the same dataset but with different strategies: one group applied $L_{\text{contra}}$ and the other does not. Both groups were evaluated on the POPE dataset. The results shown in Table 5 shows that applying adversarial contrastive loss achieved an accuracy improvement of 4.4% on both the *Adversarial* and *Popular* subsets of POPE compared to the control group. These results highlight the effectiveness of the proposed adversarial contrastive loss $L_{\text{contra}}$ to mitigate object hallucination, particularly under challenging or misleading conditions.

## A.6 DETAILED VISUALIZATION AND ANALYSIS

In this section, we provide more detailed visual examples and analysis to prove the effectiveness of ACFT. As shown in Figure 5, for samples that previously induced hallucinations, ACFT improved the cosine similarity between their image and text embeddings, and the finetuned LVLM's Grad-CAM attention maps became more tightly focused on the target objects. This, to some extent, explains the underlying mechanism of ACFT's effectiveness and confirms our initial analysis.

More concretely, we define a *semantically appropriate* attention distribution as follows: (1) If the question asks about an object that exists in the image, the attention should be focused on the target region. (2) If the question asks about an object that is absent, the attention should be more dispersed, rather than spuriously concentrating on an unrelated region (which tends to trigger hallucinations). This pattern is what Figure 5 is intended to illustrate. In Cases 1 and 4, the question refers to objects that are present. Before ACFT, the Grad-CAM maps are relatively dispersed and often miss the true target area; after ACFT, the attention becomes clearly more concentrated on the correct region, consistent with the corrected, non-hallucinated prediction. In contrast, in Cases 2 and 3, the question refers to objects that are absent. Before ACFT, the model's attention concentrates on a wrong local region and the model hallucinates the queried object there; after ACFT, the attention becomes much more distributed, and the model correctly answers that the object is not present.

We perform a quantitative analysis on the Grad-CAM maps. For each heatmap, we compute its entropy. Specifically, we first build a histogram over attention heatmap intensity values and treat this histogram as a probability distribution to calculate the Shannon entropy. We further normalize this entropy by the maximum possible entropy (the logarithm of the number of bins). Lower entropy indicates a more concentrated attention pattern, and higher entropy indicates a more distributed one. The results are consistent with the above interpretation: (1) In Cases 1 and 4 (object present), the entropy decreases by 8.7% and 2.6% after ACFT, indicating that the model's attention becomes more focused on the true object region. (2) In Cases 2 and 3 (object absent), the entropy increases by 7.4% and 6.4% after ACFT, indicating that the attention becomes more dispersed instead of over-confidently locking onto an incorrect region.

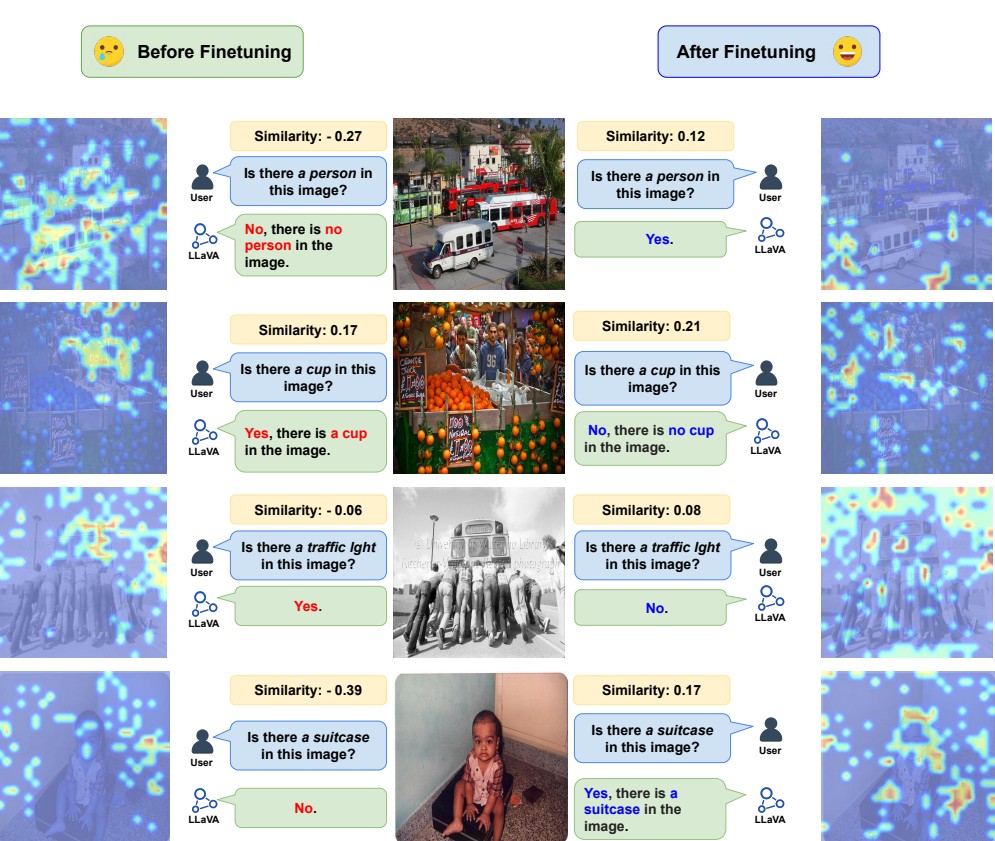

Figure 5: More detailed visualization results of our proposed method. After applying ACFT, LVLMs no longer generate hallucinated responses in these cases. ACFT improved the cosine similarity between their image and text embeddings, while the finetuned LVLM's Grad-CAM attention maps became more tightly focused on the target objects. Blue annotations indicate responses without hallucinations, while red annotations highlight parts of the responses where hallucinations occur.

## A.7  DETAILS ABOUT ADVERSARIAL EXAMPLE GENERATION IN AHAF

For the hyperparameter settings in AHAF, we select $\epsilon$ from $4/255, 8/255, 16/255, 32/255, 64/255$ and the number of iterations from $50, 100, 200, 500, 1000$, with the step size fixed at 1. With different hyperparameter settings, we performed PGD attack on 50 randomly selected images. We observe that when $\epsilon$ is too small, the number of iterations required to successfully flip the hallucinated attribute becomes very high, and manual inspection is often required to make sure the attack is successful. Conversely, when $\epsilon$ is too large, the adversarial noise becomes visually apparent, and the generated image significantly deviates from the target. To balance visual quality and attack effectiveness, we set $\epsilon$ to 16/255 and the number of iterations to 100.

Compared to two commonly used adversarial attack methods—FGSM Goodfellow et al. (2015) and CW Carlini & Wagner (2017)—PGD offers a favorable trade-off between effectiveness and computational efficiency in the AHAF task. FGSM applies a single-step perturbation, making it less robust and often yielding lower attack success rates in more challenging scenarios. On the other hand, the CW attack involves complex optimization procedures and is computationally expensive. In contrast, PGD is both simpler to implement and more scalable to large-scale datasets and models, while still delivering strong attack performance. Therefore, we adopt PGD as the primary adversarial method in our AHAF framework.

## A.8  EXPERIMENTS ON DESCRIPTION-LEVEL BENCHMARKS

we have conducted experiments on five description-level benchmarks: CHAIR Rohrbach et al. (2018), CCEval Zhai et al. (2023), AMBERA Wang et al. (2023), MMHal-Bench Sun et al. (2023), and ObjectHal Yu et al. (2024). For CHAIR, we sample 500 images from COCO 2014 val, prompt the model with "Please describe this image in detail.", and set max new token to 512. For CCEval, AMBERA, MMHal-Bench, and ObjectHal, we follow each benchmark's original evaluation setting. All experiments are conducted on LLaVA v1.5-7B Liu et al. (2023). The evaluation results are shown in Table 6. The results show that, although ACFT is trained only on short-answer data, it consistently reduced caption-level hallucination across all five benchmarks, with lower CHAIR scores and hallucination rates compared to the original model. This supports our core claim that strengthening visual perception and multimodal alignment benefits not only binary settings but also transfers to open-ended description tasks. ACFT enables the model to focus more tightly on the target region, and form more discriminative representations for present versus absent objects in the embedding space. Thus, the model is less likely to introduce nonexistent objects even when producing long, free-form captions. Although the absolute gains on caption-level benchmarks are smaller than those on binary benchmarks—unsurprising given that long-form generation is still strongly affected by language priors—we believe these results show that improving visual grounding and multimodal alignment on short-answer tasks can provide a robust backbone that generalize well to open-ended description generation.

## A.9  COMPARISON WITH POST-TRAINING BASELINES

In this section, we compare ACFT with 4 post-training baselines: (1) a SFT baseline that trains the same backbone as ACFT on the same data using a standard cross-entropy loss; (2) LLaVA-RLHF Sun et al. (2023), we directly use the model weights released by the authors; (3) OPA-DPO Yang et al. (2025b), a DPO-based method for hallucination mitigation, we use the model weights released by the authors;and (4) CHiP-DPO Fu et al. (2025), we reproduce the method using the authors' released data and training scripts. We evaluate all of these methods under the same base model(LLaVA v1.5 7B Liu et al. (2023)) and report their performance on POPE, MME-Existence, and the full MME benchmark. The results are shown in Table 7. The results indicate that, under a comparable data budget (approximately 6k samples), ACFT still achieves the best overall performance among these post-training baselines. This indicates that the gains of ACFT do not simply come from "doing more fine-tuning," but from the design of adversarially constructed positive–negative image pairs and the adversarial contrastive loss, which more directly targets the visual misalignment underlying object hallucinations.

| Benchmark | Metric | Original | ACFT |
|---|---|---|---|
| CHAIR | CHAIR$_s$ ↓ | 0.508 | **0.494** |
| | CHAIR$_i$ ↓ | 0.142 | **0.137** |
| CCEval | CHAIR$_s$ ↓ | 0.870 | **0.850** |
| | CHAIR$_i$ ↓ | 0.348 | **0.328** |
| AMBERA Generative | CHAIR ↓ | 0.112 | **0.089** |
| | Hal ↓ | 0.488 | **0.483** |
| | Cover ↑ | **0.518** | 0.513 |
| | Cog ↓ | 0.047 | **0.043** |
| AMBERA Discriminative | ACC ↑ | 0.716 | **0.729** |
| | Precision ↑ | 0.933 | **0.941** |
| | Recall ↑ | 0.617 | **0.630** |
| | F1 ↑ | 0.743 | **0.755** |
| MMHal-Bench | Score ↑ | **2.710** | 2.650 |
| | Hallucination Rate ↓ | 0.604 | **0.594** |
| ObjectHal | Response Hal ↓ | 0.568 | **0.562** |
| | Obj Hal ↓ | 0.283 | **0.277** |

Table 6: Comparison between the original model and ACFT on description-level hallucination benchmarks.

## A.10 COMPUTATIONAL COST ANALYSIS

In this section, we provide detailed analysis for both AHAF and ACFT stages.

In the AHAF stage, we construct a training set with 3,000 contrastive image samples and 3,000 normal samples. For the 3,000 contrastive samples, we run PGD on each image with 100 iterations. On a single NVIDIA A100, the average optimization time per image is about 15s, so constructing all 3,000 adversarial images takes roughly 12 GPU-hours.

In the ACFT stage, the contrastive loss only introduces only a lightweight additional computation overhead on top of standard SFT: we reuse the last-layer image and text representations to compute the adversarial contrastive loss, without extra forward passes through the backbone. In practice, for LLaVA-v1.5-7B with batch size 16 on a single A100, plain SFT training on our 6k-sample set takes 30 min 57s, while ACFT takes 32 min 21s—an overhead of about 90s, which we consider negligible. For comparison, when we run CHiP-DPO Fu et al. (2025) using the official training script on the same backbone, training requires 3 h 21 min 45 s on 4 A100 GPUs, i.e., more than 13 GPU-hours, on a dataset of comparable size. We also apply vanilla DPO to fine-tune LLaVA v1.5-7B on 6000 COCO images. The DPO training process takes 1 h 33 min 27 s on 4×A100 GPUs (about 6 GPU-hours), whereas ACFT only requires about 0.5 GPU-hours.

## A.11 EXPERIMENT ON NEW BASE MODELS

In this section, we have implemented ACFT on two recent LVLMs, Qwen2.5-VL-7B Bai et al. (2025) and InternVL-3.5-4B Wang et al. (2025) and evaluated them on POPE and MME benchmarks. Results for Qwen2.5-VL-7B are shown in Table 8. Results for InternVL-3.5-4B are shown in Table 9. For both Qwen2.5-VL and InternVL 3.5, ACFT delivers consistent gains on POPE and MME, even though the base models are already very strong. On MME Existence, both the original model and ACFT achieve 100% accuracy. This reflects that the task is relatively easy for modern VLMs, leaving no headroom for further improvement. Overall, these results support our claim that ACFT is architecture-agnostic and transfers well to recent LVLMs.

| Benchmark | Subset | Method | ACC | Precision | Recall | F1 Score |
|---|---|---|---|---|---|---|
| POPE | *Adversarial* | SFT | 0.797 | 0.743 | **0.906** | 0.817 |
| | | LLaVA-RLHF | 0.813 | 0.835 | 0.780 | 0.806 |
| | | CHiP-DPO | 0.839 | 0.923 | 0.739 | 0.821 |
| | | opadpo | 0.827 | **0.944** | 0.697 | 0.801 |
| | | ACFT | **0.841** | 0.802 | 0.905 | **0.850** |
| | *Popular* | SFT | 0.862 | 0.832 | **0.906** | 0.867 |
| | | LLaVA-RLHF | 0.847 | 0.901 | 0.780 | 0.836 |
| | | CHiP-DPO | 0.855 | 0.962 | 0.738 | 0.835 |
| | | opadpo | 0.840 | **0.975** | 0.698 | 0.813 |
| | | ACFT | **0.906** | 0.907 | 0.905 | **0.906** |
| | *Random* | SFT | 0.896 | 0.888 | **0.906** | **0.897** |
| | | LLaVA-RLHF | 0.867 | 0.943 | 0.780 | 0.854 |
| | | CHiP-DPO | 0.863 | 0.984 | 0.739 | 0.844 |
| | | opadpo | 0.845 | **0.991** | 0.696 | 0.818 |
| | | ACFT | **0.897** | 0.890 | 0.905 | **0.897** |
| MME | *Existence* | SFT | 0.950 | 0.935 | **0.967** | 0.951 |
| | | LLaVA-RLHF | 0.967 | 0.967 | **0.967** | 0.967 |
| | | CHiP-DPO | 0.967 | **1.000** | 0.933 | 0.965 |
| | | opadpo | 0.967 | **1.000** | 0.933 | 0.965 |
| | | ACFT | **0.983** | **1.000** | **0.967** | **0.983** |
| | *Whole* | SFT | 0.736 | 0.718 | 0.778 | 0.747 |
| | | LLaVA-RLHF | 0.717 | 0.800 | 0.578 | 0.671 |
| | | CHiP-DPO | 0.653 | **0.925** | 0.334 | 0.491 |
| | | opadpo | **0.753** | 0.897 | 0.572 | 0.699 |
| | | ACFT | 0.747 | 0.683 | **0.922** | **0.785** |

Table 7: Comparison of post-training baselines and ACFT on POPE and MME benchmarks.

| Benchmark | Subset | Method | ACC | Precision | Recall | F1 Score |
|---|---|---|---|---|---|---|
| POPE | *Adversarial* | original | 0.864 | **0.940** | 0.778 | 0.851 |
| | | ACFT | **0.877** | 0.897 | **0.852** | **0.874** |
| | *Popular* | original | 0.875 | **0.965** | 0.778 | 0.861 |
| | | ACFT | **0.900** | 0.942 | **0.852** | **0.895** |
| | *Random* | original | 0.884 | **0.987** | 0.778 | 0.870 |
| | | ACFT | **0.916** | 0.976 | **0.852** | **0.910** |
| MME | *Existence* | original | **1.000** | **1.000** | **1.000** | **1.000** |
| | | ACFT | **1.000** | **1.000** | **1.000** | **1.000** |
| | *Whole* | original | 0.870 | 0.836 | **0.920** | **0.878** |
| | | ACFT | **0.874** | **0.928** | 0.810 | 0.865 |

Table 8: Comparison between the original Qwen2.5 VL 7B model and ACFT model on POPE and MME benchmarks.

| Benchmark | Subset | Method | ACC | Precision | Recall | F1 Score |
|-----------|--------|--------|-----|-----------|--------|----------|
| POPE | *Adversarial* | original | 0.863 | 0.835 | **0.905** | 0.869 |
| | | ACFT | **0.876** | **0.864** | 0.893 | **0.878** |
| | *Popular* | original | 0.899 | 0.894 | **0.905** | 0.899 |
| | | ACFT | **0.912** | **0.927** | 0.895 | **0.911** |
| | *Random* | original | 0.933 | **0.969** | 0.845 | 0.930 |
| | | ACFT | **0.937** | 0.966 | **0.905** | **0.935** |
| MME | *Existence* | original | **1.000** | **1.000** | **1.000** | **1.000** |
| | | ACFT | **1.000** | **1.000** | **1.000** | **1.000** |
| | *Whole* | original | 0.859 | **0.913** | 0.795 | 0.850 |
| | | ACFT | **0.862** | 0.894 | **0.821** | **0.856** |

Table 9: Comparison between the original InternVL 3.5 4B model and ACFT model on POPE and MME benchmarks.