# OpenReview forum: "Mitigating Object Hallucination in Large Vision-Language Models through Adversarial Contrastive Finetuning"
_ICLR.cc/2026/Conference — Submitted to ICLR 2026_

### Official Review · Reviewer_iJhQ · 2025-10-16

**Soundness:** 1
**Presentation:** 2
**Contribution:** 2
**Rating:** 4
**Confidence:** 4

**Summary:**

This paper proposes Adversarial Contrastive Fine-tuning (ACFT) to mitigate object hallucination in LVLMs.
The method introduces Adversarial Hallucination Attribute Flipping (AHAF) to automatically generate aligned positive–negative image pairs, followed by contrastive fine-tuning that maximizes text–image embedding similarity for positives and minimizes it for negatives.

**Strengths:**

1. The logical organization of the paper is clear and coherent, making it easy to follow.
2. The motivation is well-articulated and grounded in empirical observation, effectively linking visual perturbations to hallucination behavior.

**Weaknesses:**

1. Limited benchmark coverage.

(a) The evaluation focuses only on binary benchmarks (POPE, MME). The paper should also consider generative hallucination evaluations such as AMBERA [3], MMHal-Bench [4], and ObjectHal [5], which assess open-ended generation quality rather than yes/no predictions.

(b) Without these benchmarks, it is difficult to verify whether ACFT generalizes beyond binary tasks.

2. Lack of fair comparison with fine-tuning baselines.

(a) The paper omits DPO-based hallucination mitigation approaches that achieve strong results with very limited data, e.g., CHiP-DPO [1], On-Policy DPO [2].

(b) These methods typically use only around 5k preference pairs yet reach or exceed state-of-the-art results, which questions the efficiency advantage claimed by ACFT.

3. Data efficiency and computational cost unclear.

(a) The paper mentions using “0.9% of COCO” but does not specify the actual number of samples or the corresponding GPU hours.

(b) Without a comparable-scale baseline (e.g., DPO fine-tuning with 0.9% of COCO pairs), the efficiency claim is not well supported.

4. Reproducibility concerns on VCD results.

(a) The reported VCD results on LLaVA-1.5-7B differ significantly from the original paper [6].

(b) The original work reported around +2% Acc and +4% F1 gains under adversarial settings, whereas this paper shows lower-than-baseline results **(w/ VCD < w/o VCD)**, suggesting inconsistent setups that may compromise the fairness and reproducibility of comparison.

5. Conceptual limitation in contrastive design.

(a) Positive/negative pairs are created only in the image domain, while text-side perturbations or multimodal feature contrasts are not explored.

(b) It remains unclear whether similar or better robustness could be achieved via token-level or text-level contrastive learning.


[1] Chip: Cross-modal hierarchical direct preference optimization for multimodal llms, ICLR 2025.

[2] Mitigating hallucinations in large vision-language models via dpo: On-policy data hold the key, CVPR 2025.

[3] Amber: An llm-free multi-dimensional benchmark for mllms hallucination evaluation,  2023.

[4] Aligning large multimodal models with factually augmented rlhf, ACL 2024.

[5] Rlhf-v: Towards trustworthy mllms via behavior alignment from fine-grained correctional human feedback, CVPR 2024.

[6] Mitigating object hallucinations in large vision-language models through visual contrastive decoding, CVPR 2024

**Questions:**

Please refer to the issues discussed in Weaknesses.
In particular, I would like the authors to clarify the following:

(a) Whether the experimental setup is fair and consistent, especially regarding the reported VCD [6] results.
(b) Whether, under a comparable fine-tuning data scale, the proposed method can achieve state-of-the-art performance similar to DPO-based approaches [1–2].
(c) Whether the method can maintain its advantage on generative hallucination benchmarks, such as AMBERA [3], MMHal-Bench [4], and ObjectHal [5].

If these points can be convincingly addressed, I would consider raising my overall rating.

---

> ### Author Response · Authors · 2025-11-22
> **Official Comment by Authors (1/4)**
>
> **Weakness 1: Limited benchmark coverage**
>
> We thank the reviewer for the insightful comment.
>
> Firstly, we want to clarify **why we focused on binary benchmarks**. As discussed in the Introduction section, most existing works analyze object hallucination from the text side and attribute hallucination mainly to over-reliance on language priors, while the visual dimension remains relatively underexplored. This motivates us to study and mitigate object hallucination from the image-prior perspective, which is different from most previous works. Binary “Yes/No” benchmarks are particularly suitable for this purpose: their short answers reduce the influence of language priors. These benchmarks help us isolate hallucinations that arise from visual features or multimodal misalignment rather than from long-range text generation. Our work does not conflict with previous studies; instead, they approach the problem from two different perspectives and are actually complementary to each other.
>
> We fully agree with the reviewer that showing generalization to open-ended, caption-level hallucination is important. Thus, we have conducted experiments on five description-level benchmarks: **CHAIR, CCEval, AMBERA, MMHal-Bench, and ObjectHal**. For CHAIR, we sample 500 images from COCO 2014 val, prompt the model with “Please describe this image in detail.”, and set max_new_token to 512. For CCEval, AMBERA, MMHal-Bench, and ObjectHal, we follow each benchmark’s original evaluation setting.  All experiments are conducted on LLaVA v1.5-7B.
>
> | **Benchmark**            | **Metric**                    | **Original** | **ACFT**       |
> |--------------------------|-------------------------------|-------------:|---------------:|
> | **CHAIR**                | CHAIR$_s$ ${\downarrow}$      | 0.508        | **0.494**      |
> |                          | CHAIR$_i$ ${\downarrow}$      | 0.142        | **0.137**      |
> | **CCEval**               | CHAIR$_s$ ${\downarrow}$      | 0.870        | **0.850**      |
> |                          | CHAIR$_i$ ${\downarrow}$      | 0.348        | **0.328**      |
> | **AMBERA Generative**    | CHAIR ${\downarrow}$          | 0.112        | **0.089**      |
> |                          | Hal ${\downarrow}$            | 0.488        | **0.483**      |
> |                          | Cover ${\uparrow}$            | **0.518**    | 0.513          |
> |                          | Cog ${\downarrow}$            | 0.047        | **0.043**      |
> | **AMBERA Discriminative**| ACC ${\uparrow}$              | 0.716        | **0.729**      |
> |                          | Precision ${\uparrow}$        | 0.933        | **0.941**      |
> |                          | Recall ${\uparrow}$           | 0.617        | **0.630**      |
> |                          | F1 ${\uparrow}$               | 0.743        | **0.755**      |
> | **MMHal-Bench**          | Score ${\uparrow}$            | **2.710**    | 2.650          |
> |                          | Hallucination Rate ${\downarrow}$ | 0.604   | **0.594**      |
> | **ObjectHal**            | Response Hal ${\downarrow}$   | 0.568        | **0.562**      |
> |                          | Obj Hal ${\downarrow}$        | 0.283        | **0.277**      |
>
> The results show that, although ACFT is trained **only** on short-answer data, it consistently reduced caption-level hallucination **across all five benchmarks**, with lower CHAIR scores and hallucination rates compared to the original model. This supports our core claim that strengthening visual perception and multimodal alignment **benefits not only binary settings but also transfers to open-ended description tasks**. ACFT enables the model to focus more tightly on the target region, and form more discriminative representations for present versus absent objects in the embedding space. Thus, the model  is less likely to introduce nonexistent objects even when producing long, free-form captions. Although the absolute gains on caption-level benchmarks are smaller than those on binary benchmarks, it is unsurprising given that long-form generation is still strongly affected by language priors. We believe these results show that improving visual grounding and multimodal alignment on short-answer tasks can provide a robust backbone that generalizes well to open-ended description generation.
>
> We have included these new caption-level results in the revised paper in Appendix A.8.

---

> > ### Author Response · Authors · 2025-11-22
> > **Official Comment by Authors (2/4)**
> >
> > **Weakness 2: Lack of fair comparison with fine-tuning baselines**
> >
> > Thank you for raising this important concern.
> >
> > To address this, we have added 4 post-training baselines: (1) a **SFT baseline** that trains the same backbone as ACFT on the same data using a standard cross-entropy loss; (2) **LLaVA-RLHF**, we directly use the model weights released by the authors; (3) **OPA-DPO**, a DPO-based method for hallucination mitigation, we use the model weights released by the authors; and (4) **CHiP-DPO**, we reproduce the method using the authors’ released data and training scripts. We evaluate all of these methods under the same base model(LlaVA v1.5 7B) and report their performance on POPE, MME-Existence, and the full MME benchmark. The results are shown below.
> >
> > | **Benchmark** | **Subset**        | **Method**    | **ACC** | **Precision** | **Recall** | **F1 Score** |
> > |---------------|-------------------|---------------|--------:|-------------:|----------:|------------:|
> > | **POPE**      | *Adversarial*     | SFT           | 0.797   | 0.743        | **0.906** | 0.817       |
> > |               |                   | LLaVA-RLHF    | 0.813   | 0.835        | 0.780     | 0.806       |
> > |               |                   | CHiP-DPO      | 0.839   | 0.923        | 0.739     | 0.821       |
> > |               |                   | opadpo        | 0.827   | **0.944**    | 0.697     | 0.801       |
> > |               |                   | **ACFT**      | **0.841** | 0.802      | 0.905     | **0.850**   |
> > | **POPE**      | *Popular*         | SFT           | 0.862   | 0.832        | **0.906** | 0.867       |
> > |               |                   | LLaVA-RLHF    | 0.847   | 0.901        | 0.780     | 0.836       |
> > |               |                   | CHiP-DPO      | 0.855   | 0.962        | 0.738     | 0.835       |
> > |               |                   | opadpo        | 0.840   | **0.975**    | 0.698     | 0.813       |
> > |               |                   | **ACFT**      | **0.906** | 0.907      | 0.905     | **0.906**   |
> > | **POPE**      | *Random*          | SFT           | 0.896   | 0.888        | **0.906** | **0.897**   |
> > |               |                   | LLaVA-RLHF    | 0.867   | 0.943        | 0.780     | 0.854       |
> > |               |                   | CHiP-DPO      | 0.863   | 0.984        | 0.739     | 0.844       |
> > |               |                   | opadpo        | 0.845   | **0.991**    | 0.696     | 0.818       |
> > |               |                   | **ACFT**      | **0.897** | 0.890      | 0.905     | **0.897**   |
> > | **MME**       | *Existence*       | SFT           | 0.950   | 0.935        | **0.967** | 0.951       |
> > |               |                   | LLaVA-RLHF    | 0.967   | 0.967        | **0.967** | 0.967       |
> > |               |                   | CHiP-DPO      | 0.967   | **1.000**    | 0.933     | 0.965       |
> > |               |                   | opadpo        | 0.967   | **1.000**    | 0.933     | 0.965       |
> > |               |                   | **ACFT**      | **0.983** | **1.000**  | **0.967** | **0.983**   |
> > | **MME**       | *Whole*           | SFT           | 0.736   | 0.718        | 0.778     | 0.747       |
> > |               |                   | LLaVA-RLHF    | 0.717   | 0.800        | 0.578     | 0.671       |
> > |               |                   | CHiP-DPO      | 0.653   | **0.925**    | 0.334     | 0.491       |
> > |               |                   | opadpo        | **0.753** | 0.897      | 0.572     | 0.699       |
> > |               |                   | **ACFT**      | 0.747   | 0.683        | **0.922** | **0.785**   |
> >
> > The results show that, under a comparable data budget (approximately 6k samples), ACFT still outperforms these post-training baselines. This indicates that **the gains of ACFT do not simply come from “doing more fine-tuning,” but from the design of adversarially constructed positive–negative image pairs and the adversarial contrastive loss**, which more directly targets the visual misalignment underlying object hallucinations.
> >
> > We have added these post-training baselines and their results to our revised paper in Appendix A.9.

---

> > > ### Author Response · Authors · 2025-11-22
> > > **Official Comment by Authors (3/4)**
> > >
> > > **Weakness 3: Data efficiency and computational cost unclear.**
> > >
> > > We thank the reviewer for pointing out this concern. Here we provide detailed efficiency analysis for ACFT.
> > >
> > > Our training set contains two parts: (1) 3,000 contrastive samples constructed by AHAF, and (2) 3,000 normal VQA samples to avoid overfitting to binary yes/no answering format.
> > >
> > > During fine-tuning, ACFT introduces **only a lightweight additional computation overhead** on top of standard SFT: we reuse the last-layer image and text representations to compute the adversarial contrastive loss, without extra forward passes through the backbone. In practice, for LLaVA-v1.5-7B with batch size 16 on a single A100, plain SFT training on our 6k-sample set takes **30 min 57s**, while ACFT takes **32 min 21s**. ACFT only brings an overhead of about 90s, which we consider negligible. For comparison, when we run CHiP-DPO using the official training script on the same backbone, training requires 3 h 21 min 45s on 4 A100 GPUs, i.e., more than **13 GPU-hours**, on a dataset of comparable scale.
> > >
> > > Following the reviewer’s suggestion, we also apply vanilla DPO to fine-tune LLaVA v1.5-7B on 6000 COCO images. For each example, we construct rejected responses by flipping the ground-truth label (yes → no, no → yes). The DPO training process takes 1 h 33 min 27 s on 4×A100 GPUs (about  **6 GPU-hours**), whereas ACFT only requires about **0.5 GPU-hours**. Moreover, as shown in the results, ACFT achieves better performance on POPE. The results suggest that text-only DPO is insufficient and highlight both the efficiency and effectiveness of the ACFT design.
> > >
> > > | Subset   | Method | Accuracy | Precision | Recall | F1 score |
> > > |----------|--------|---------:|----------:|-------:|---------:|
> > > | adv      | DPO    | 0.830    | 0.792     | 0.895  | 0.840    |
> > > |          | ACFT   | **0.841**| **0.802** | **0.905** | **0.850** |
> > > | popular  | DPO    | 0.890    | 0.885     | 0.895  | 0.890    |
> > > |          | ACFT   | **0.906**| **0.907** | **0.905** | **0.906** |
> > > | random   | DPO    | 0.892    | **0.907** | 0.874  | 0.890    |
> > > |          | ACFT   | **0.897**| 0.890     | **0.905** | **0.897** |
> > >
> > > We have included computational cost analysis in Appendix A.10 in our revised paper.
> > >
> > > **Weakness 4: Reproducibility concerns on VCD results.**
> > >
> > > We thank the reviewer for raising this concern.
> > >
> > > Firstly, we would like to clarify that we did implement VCD following its official repository, and the original numbers in our submission reflected the results obtained from this local re-implementation. We agree that the discrepancy with the original paper is non-trivial and have carefully investigated possible causes:
> > >
> > > - To ensure a fair comparison across all baselines, our initial experiments used **a single inference codebase and a unified set of decoding hyperparameters for all methods**, and applied VCD only as a change in the decoding strategy. This deviates from the reference evaluation script used in the original VCD paper, and differences in sampling/decoding settings (e.g., temperature, top-p, max length) may affect the results.
> > >
> > > - For compatibility with later baselines like VTI, we used **a more recent environment(including newer versions of LLaVA and the transformers library) than the one used in the original VCD paper**. Such version drift can also cause changes in benchmark scores.
> > >
> > > We acknowledge that, in our initial submission, we made some changes for each baseline’s own recommended setup to ensure all methods are under the same evaluation setting, which may have unintentionally penalized some methods, such as VCD. Thus, we have reproduced all the baseline results again, this time strictly following each method’s released experimental setup and selecting the optimal results for each baseline. **In this way, we got the updated VCD results that align with the original paper**, and we have updated the results in the revised manuscript.

---

> > > > ### Author Response · Authors · 2025-11-22
> > > > **Official Comment by Authors (4/4)**
> > > >
> > > > **Weakness 5: Conceptual limitation in contrastive design.**
> > > >
> > > > Thank you for bringing out this concern.
> > > >
> > > > Firstly, we want to stress that **constructing positive/negative pairs in the image domain is not an arbitrary design choice but a core part of our contribution and problem formulation**. As discussed in the Introduction section, most existing approaches analyze object hallucination from the text side and primarily attribute it to over-reliance on language priors, while the visual dimension has been relatively underexplored. This motivates us to study and mitigate object hallucination through image-domain contrastive learning, rather than proposing yet another text-prior method.
> > > >
> > > > That said, we fully agree with the reviewer that it is important to understand how token-level or text-level contrastive learning compares to our image-domain design. To investigate this, we implement **a text-level contrastive learning method using the same training data as ACFT**. For each question, we construct contrastive text pairs by flipping the ground-truth label (“yes” → “no”, “no” → “yes”), and perform contrastive learning at the text level while keeping the images fixed. We then fine-tune LLaVA v1.5-7B with this text-level contrastive objective and evaluate on the POPE benchmark.
> > > >
> > > > | Subset   | Method      | Accuracy | Precision | Recall | F1 score |
> > > > |----------|-------------|---------:|----------:|-------:|---------:|
> > > > | adv      | Text Level  | 0.805    | 0.762     | 0.887  | 0.820    |
> > > > |          | Image Level | **0.841**| **0.802** | **0.905** | **0.850** |
> > > > | popular  | Text Level  | 0.883    | 0.889     | 0.887  | 0.888    |
> > > > |          | Image Level | **0.906**| **0.907** | **0.905** | **0.906** |
> > > > | random   | Text Level  | 0.851    | 0.847     | 0.857  | 0.852    |
> > > > |          | Image Level | **0.897**| **0.890** | **0.905** | **0.897** |
> > > >
> > > > As shown in the table, text-level contrastive learning yields only modest gains and **consistently underperforms our image-domain ACFT across all POPE subsets**. This further proves the effectiveness and necessity of our image-domain contrastive design.

---

> > > > > ### Comment · Reviewer_iJhQ · 2025-11-25
> > > > >
> > > > > I thank the authors for their detailed responses and the significant effort put into the additional experiments, including the inclusion of DPO-based baselines, efficiency analysis, and the correction of the VCD results.
> > > > >
> > > > > While the rebuttal has addressed several of my concerns (particularly regarding fair comparison and efficiency), I have decided to maintain my original score. My rationale is as follows:
> > > > >
> > > > > 1. Marginal Improvements on Generative Benchmarks: Although the authors included open-ended benchmarks (e.g., CHAIR, AMBER) as requested, the performance gains are quite marginal (e.g., CHAIR metric improving only from 0.142 to 0.137). This reinforces my concern that while the method is highly effective for binary discrimination tasks (like POPE), its ability to generalize to more complex, open-ended captioning tasks—which are crucial for real-world LVLM applications—remains limited.
> > > > >
> > > > > 2. Scope of Contribution: I acknowledge the method's computational efficiency and its logic in targeting visual misalignment. However, given the limited improvement on open-ended generation, the method appears somewhat over-optimized for binary/discriminative settings.
> > > > >
> > > > > In summary, while the paper presents a sound and efficient method, the limited impact on broader generative tasks prevents me from raising the score to an acceptance level at this stage.

---

> > > > > > ### Author Response · Authors · 2025-11-26
> > > > > >
> > > > > > We thank the reviewer for acknowledging the additional content in our rebuttal.
> > > > > >
> > > > > > Regarding the concern that ACFT yields only marginal improvements on generative benchmarks, we would like to clarify the following points:
> > > > > >
> > > > > > Firstly, we want to stress that **hallucination in LVLMs is a multifaceted problem, and no single method can address all forms of hallucination across all tasks** [1,2]. ACFT focuses on image-prior induced hallucination, which is a common but relatively under-explored cause of hallucination. While binary questions may appear trivial at the language level, the underlying visual decision behind these questions can be quite challenging.
> > > > > >
> > > > > > Secondly, we would like to clarify that our goal is not to claim that image-prior methods such as ACFT are universally superior to language-prior approaches. Rather, **they tackle hallucination from complementary perspectives**: language-prior methods primarily operate at the decoding stage to reduce reliance on text priors, whereas ACFT focuses on enhancing visual perception and multimodal alignment in the training stage. In this sense, ACFT is designed to be **synergistic** with language-prior approaches, not a replacement.
> > > > > >
> > > > > > To prove this, we have added experiments combining *ACFT* with a language-prior method, *OPERA*. We evaluate four variants: (i) the original model, (ii) OPERA, (iii) ACFT, and (iv) ACFT + OPERA, on both a binary benchmark (*POPE*) and a generative benchmark (*CHAIR*).
> > > > > >
> > > > > > Results on generative benchmark:
> > > > > >
> > > > > > | Method      | CHAIR_s | CHAIR_i |
> > > > > > |------------|---------|---------|
> > > > > > | *original*   | 0.508   | 0.142   |
> > > > > > | *ACFT*       | 0.494   | 0.137   |
> > > > > > | *OPERA*      | 0.499   | 0.125   |
> > > > > > | *OPERA+ACFT* | **0.487**   | **0.121**   |
> > > > > >
> > > > > > Results on binary benchmark:
> > > > > >
> > > > > > |**Benchmark**|**Subset**|**Method**|**ACC**|**Precision**|**Recall**|**F1 Score**|
> > > > > > |-------------|----------|---------:|-----:|-----------:|--------:|----------:|
> > > > > > |**POPE**|*Adversarial*|Original|0.779|0.721|**0.911**|0.805|
> > > > > > |||OPERA|0.798|0.787|0.816|0.802|
> > > > > > |||ACFT|0.841|0.802|0.905|0.850|
> > > > > > |||**OPERA+ACFT**|**0.855**|**0.833**|0.887|**0.860**|
> > > > > > |**POPE**|*Popular*|Original|0.862|0.832|0.905|0.867|
> > > > > > |||OPERA|0.886|0.847|**0.940**|0.891|
> > > > > > |||ACFT|0.906|0.907|0.905|0.906|
> > > > > > |||**OPERA+ACFT**|**0.918**|**0.916**|0.920|**0.918**|
> > > > > > |**POPE**|*Random*|Original|0.885|0.867|**0.910**|0.888|
> > > > > > |||OPERA|0.878|**0.918**|0.831|0.873|
> > > > > > |||ACFT|0.897|0.890|0.905|0.897|
> > > > > > |||**OPERA+ACFT**|**0.910**|0.910|0.909|**0.910**|
> > > > > >
> > > > > > From the results, we have three key observations:
> > > > > >
> > > > > > - **Language-prior methods also have limitations in certain settings.** On binary tasks like POPE, OPERA alone brings only modest gains. In contrast, ACFT, which directly improves visual alignment, brings much larger improvements in this setting.
> > > > > >
> > > > > > - **ACFT + OPERA achieves the best performance on both binary and generative benchmarks.** This further supports our point that strengthening visual perception (ACFT) and reducing the effect of language priors (OPERA) address different failure modes and reinforce each other. ACFT can mitigate hallucinations that language-prior methods fail to address in both settings.
> > > > > >
> > > > > > - **ACFT is a useful complementary building block.** Instead of proposing yet another language-side decoding strategy, our contribution is to introduce a visual-prior training method that can be combined with any existing language-prior pipelines and yield consistent improvements across diverse settings, including open-ended captioning.
> > > > > >
> > > > > > In conclusion, we acknowledge that the absolute improvements achieved by ACFT on open-ended datasets are smaller than those on POPE. However, open-ended captioning benchmarks are known to be **noisier and more saturated**, so even modest improvement can be meaningful in practice. More importantly, the additional OPERA+ACFT results show that once we integrate ACFT with a language-prior method, the model achieves **significant improvement** on both discriminative and generative benchmarks.[3] This indicates that ACFT can mitigate hallucinations that language-prior methods fail to address. Thus, ACFT is not “over-optimized” for binary tasks, but instead **provides a robust visual foundation that other mitigation techniques can effectively build upon**.
> > > > > >
> > > > > > [1] Sun, Yaqi, et al. "Exploring Causes and Mitigation of Hallucinations in Large Vision Language Models." arXiv preprint arXiv:2502.16842 (2025).
> > > > > >
> > > > > > [2] Bai, Zechen et al. “Hallucination of Multimodal Large Language Models: A Survey.” ArXiv abs/2404.18930 (2024): n. pag.
> > > > > >
> > > > > > [3] Huang, Qidong et al. “OPERA: Alleviating Hallucination in Multi-Modal Large Language Models via Over-Trust Penalty and Retrospection-Allocation.” 2024 IEEE/CVF Conference on Computer Vision and Pattern Recognition (CVPR) (2023): 13418-13427.

---

### Official Review · Reviewer_TbAL · 2025-10-26

**Soundness:** 4
**Presentation:** 3
**Contribution:** 3
**Rating:** 8
**Confidence:** 4

**Summary:**

This paper addresses the problem of object hallucination in Large Vision-Language Models. It identifies visual-textual misalignment and erroneous visual feature extraction as key causes, particularly in short-text outputs. To mitigate this, the authors propose a novel Adversarial Contrastive Fine-Tuning (ACFT) method, which uses adversarially generated image pairs to enhance visual-textual alignment. Experimental results on benchmarks like POPE and MME show that ACFT achieves state-of-the-art performance in reducing hallucinations without compromising general visual understanding.

**Strengths:**

The paper demonstrates strong originality by creatively combining adversarial training with contrastive learning in a novel framework (AHAF+ACFT). It formulates the hallucination problem from an under-explored image-prior perspective, specifically targeting the short-text-output scenario, which provides a fresh alternative to dominant language-prior approaches.

The work is technically sound, evidenced by rigorous quantitative analysis (cosine similarity, benchmark results) and qualitative validation (Smooth Grad-CAM). Its significance is high: ACFT achieves SOTA performance while being highly efficient (using only 0.9% of COCO data), making robust hallucination mitigation more practical for real-world applications without added inference cost.

**Weaknesses:**

* ***Scope Limitation and Generalizability***.
The paper's exclusive focus on short-text outputs (Yes/No QA) is a significant limitation. While strategically chosen to argue for the image-prior perspective, it leaves the method's efficacy in more common, open-ended long-text generation tasks (e.g., captioning, detailed description) completely unverified. The claimed superiority over "language-prior" methods is thus narrowly contextualized, and the core contribution would be substantially strengthened by demonstrating that ACFT's benefits transfer to these standard hallucination evaluation settings.

* ***Lack of Cross-Dataset Validation***
The experimental validation is confined to two benchmarks (POPE and MME) which, despite their relevance, are structurally similar (both are VQA-style). The method's robustness would be more convincingly demonstrated by evaluating on a dataset featuring a different type of short-answer task (e.g., object counting in a non-VQA format) or from a different data distribution, to ensure the improvements are not overfitted to the specific QA format of the primary benchmarks.

**Questions:**

*   **Q1:** The AHAF method uses PGD to create negative samples. To what extent are the learned robust features specific to defending against PGD-style, norm-bound perturbations? Have you explored if the improvements hold against other types of distribution shifts or semantic perturbations that might more naturally cause hallucinations (e.g., occlusions, unusual object contexts)?
*   **Q2:** In line 42, what is "hallucination heads"?

---

> ### Author Response · Authors · 2025-11-22
> **Official Comment by Authors (1/2)**
>
> **Weakness 1: Lack of evaluation on more common, open-ended long-text generation tasks**
>
> We thank the reviewer for the insightful comment.
>
> Firstly, we want to clarify that our intention is not to claim that image-prior methods such as ACFT are universally superior to “language-prior” approaches. Rather, they tackle hallucination from complementary perspectives: language-prior methods primarily act at the decoding stage to reduce reliance on text priors, whereas ACFT focuses on enhancing the model’s visual perception and multimodal alignment in the training stage. In this sense, the two are naturally synergistic: ACFT improves the backbone’s visual representations and alignment, while language-prior methods can still be applied on top as inference-time or latent-space controllers.
>
> We fully agree with the reviewer that showing generalization to open-ended, caption-level hallucination is important. Thus, we have conducted experiments on five description-level benchmarks: **CHAIR, CCEval, AMBERA, MMHal-Bench, and ObjectHal**. For CHAIR, we sample 500 images from COCO 2014 val, prompt the model with “Please describe this image in detail.”, and set max_new_token to 512. For CCEval, AMBERA, MMHal-Bench, and ObjectHal, we follow each benchmark’s original evaluation setting.  All experiments are conducted on LLaVA v1.5-7B.
> | **Benchmark**            | **Metric**                    | **Original** | **ACFT**       |
> |--------------------------|-------------------------------|-------------:|---------------:|
> | **CHAIR**                | CHAIR$_s$ ${\downarrow}$      | 0.508        | **0.494**      |
> |                          | CHAIR$_i$ ${\downarrow}$      | 0.142        | **0.137**      |
> | **CCEval**               | CHAIR$_s$ ${\downarrow}$      | 0.870        | **0.850**      |
> |                          | CHAIR$_i$ ${\downarrow}$      | 0.348        | **0.328**      |
> | **AMBERA Generative**    | CHAIR ${\downarrow}$          | 0.112        | **0.089**      |
> |                          | Hal ${\downarrow}$            | 0.488        | **0.483**      |
> |                          | Cover ${\uparrow}$            | **0.518**    | 0.513          |
> |                          | Cog ${\downarrow}$            | 0.047        | **0.043**      |
> | **AMBERA Discriminative**| ACC ${\uparrow}$              | 0.716        | **0.729**      |
> |                          | Precision ${\uparrow}$        | 0.933        | **0.941**      |
> |                          | Recall ${\uparrow}$           | 0.617        | **0.630**      |
> |                          | F1 ${\uparrow}$               | 0.743        | **0.755**      |
> | **MMHal-Bench**          | Score ${\uparrow}$            | **2.710**    | 2.650          |
> |                          | Hallucination Rate ${\downarrow}$ | 0.604   | **0.594**      |
> | **ObjectHal**            | Response Hal ${\downarrow}$   | 0.568        | **0.562**      |
> |                          | Obj Hal ${\downarrow}$        | 0.283        | **0.277**      |
>
> The results show that, although ACFT is trained **only** on short-answer data, it consistently reduced caption-level hallucination **across all five benchmarks**, with lower CHAIR scores and hallucination rates compared to the original model. This supports our core claim that strengthening visual perception and multimodal alignment **benefits not only binary settings but also transfers to open-ended description tasks**. ACFT enables the model to focus more tightly on the target region, and form more discriminative representations for present versus absent objects in the embedding space. Thus, the model  is less likely to introduce nonexistent objects even when producing long, free-form captions. Although the absolute gains on caption-level benchmarks are smaller than those on binary benchmarks, it is unsurprising given that long-form generation is still strongly affected by language priors. We believe these results show that improving visual grounding and multimodal alignment on short-answer tasks can provide a robust backbone that generalizes well to open-ended description generation.
>
> We have included these new caption-level results in the revised paper in Appendix A.8.

---

> > ### Author Response · Authors · 2025-11-22
> > **Official Comment by Authors (2/2)**
> >
> > **Weakness 2: Lack of cross-dataset validation**
> >
> > We thank the reviewer for this valuable suggestion.
> >
> > We fully agree that evaluating on datasets with diverse task formats and distributions is important. In response to **Weakness 1**, we have extended our evaluation to several open-ended, caption-level benchmarks, whose task formats and data distributions greatly differ from POPE and MME, and ACFT still has consistent improvements under this setting. Following your suggestion, we also evaluated our method on a different type of short-answer task, **CountQABench**, an object counting benchmark.
> >
> > | CountQABench | ACC |
> > |------|-------------|
> > | original  |0.124   |
> > | ACFT   | **0.132**   |
> >
> > We also observe a slight improvement in CountQABench, indicating that the robust visual features learned by ACFT are not overfit to the binary QA format. As for why the gain is smaller than in the binary setting, we would like to emphasize that ACFT is specifically targeted at **object hallucination**, which is defined as "the model incorrectly predicting **the existence** of an object."[1] To our knowledge, most short-answer style benchmarks for evaluating object hallucination are organized in a binary “Yes/No” format. In contrast, object counting tasks evaluate numerical accuracy (i.e., how many objects are present) and, while related, are not strictly the same phenomenon as object hallucination, nor are they the primary focus of our method.
> >
> > [1] Li, Yifan, et al. "Evaluating object hallucination in large vision-language models." arXiv preprint arXiv:2305.10355 (2023).
> >
> > **Question 1: Do the improvements obtained by ACFT hold against other types of distribution shifts or semantic perturbations?**
> >
> > Thank you for raising this question.
> >
> > Firstly, although AHAF uses PGD to generate contrastive samples during training, **all our evaluations (POPE and MME) are conducted on natural images without any adversarial perturbations**. These benchmarks contain diverse real-world variations (viewpoint, clutter, illumination, background context), and ACFT consistently outperforms the baseline models on them.
> >
> > Secondly, as discussed in our responses to **Weakness 1 and 2**, we further test ACFT under distribution shifts in both data and task format. We test on several benchmarks with different question styles and data distributions, and ACFT still works well.
> >
> > Thirdly, following your suggestion, we test the **robustness of ACFT to semantic perturbations via occlusion**. We randomly sample 1,000 images from COCO, select target objects, and apply occlusion by masking out the bounding box of the target object. We then construct questions of the form “Is there a {target} in the image?” and evaluate the original model and the ACFT-finetuned model on these occluded images. The results shown in the table below indicate that ACFT still maintains a clear advantage over the baseline.
> > | Metric | ACC |
> > |------|-------------|
> > | original  |0.876   |
> > | ACFT   | **0.932**   |
> >
> > Taken together, these results prove that the robust visual features learned via PGD-based AHAF are **not specific to defending against norm-bounded perturbations, but also transfer to natural images with diverse distributions and semantic perturbations**.
> >
> > **Question 2: What is "hallucination heads"?**
> >
> > “Hallucination head” is a concept introduced in the reference [1]. The authors define hallucination heads as **attention heads in the Transformer whose activations causally contribute the most to generating hallucinated words**, i.e., tokens that describe objects not supported by the image. They identify these heads by measuring how much ablating each head reduces the probability of hallucination tokens compared to non-hallucination tokens, and refer to those with high contrastive influence as hallucination heads. Empirically, these heads mostly appear in middle or deep layers and tend to over-attend to text tokens rather than image tokens.
> >
> > [1] https://openreview.net/forum?id=Bjq4W7P2Us

---

> > > ### Comment · Reviewer_TbAL · 2025-11-27
> > >
> > > After reading the rebuttal, I maintain my original assessment that this paper is, at the very least, a clear contribution.
> > >
> > > To AC: Initially, I was genuinely torn between assigning a score of 6 or 8. However, I believe the authors’ proposed analytical perspective—specifically, the image-prior viewpoint—is notably novel, especially when compared to work in the same period on hallucination. Moreover, the results are convincing. I have also reviewed the concerns raised by other reviewers; if the authors adequately address all these limitations especially experiments in the revised manuscript, the paper could become a solid one.

---

> > > > ### Author Response · Authors · 2025-11-27
> > > >
> > > > Thank you very much for your thoughtful assessment and for recognizing that our work presents “at the very least, a clear contribution”. We are truly grateful that you view our image-prior analytical perspective as “notably novel compared to work in the same period on hallucination,” and that you find our results “convincing”. Targeting image-prior induced hallucination was indeed our main motivation and major contribution, and we are very glad that this came through in your reading.
> > > >
> > > > We also sincerely appreciate your careful consideration of the other reviewers’ concerns. In the revised manuscript, we have made a concerted effort to address these limitations. We have added further experiments and provided more detailed analysis both in the rebuttal and the revised manuscript, in line with the suggestions from all reviewers.

---

### Official Review · Reviewer_xjhn · 2025-10-27

**Soundness:** 3
**Presentation:** 3
**Contribution:** 2
**Rating:** 4
**Confidence:** 3

**Summary:**

This paper investigates the causes of object hallucination inVLM, proposing that it stems from the misalignment between visual and textual features through quantitative analysis usingcosine similarity between image and text embeddings and qualitative analysis using Grad-CAM. Based on this diagnosis, the paper introduces Adversarial Contrastive Finetuning, a method that first generates aligned positive and negative image pairs using an Adversarial Hallucination Attribute Flipping technique. It then uses these pairs for contrastive finetuning to improve visual-textual alignment . Experiments on LLaVA-1.5-7B and MiniGPT-4-13B improve performance on the POPE and MME-Existence benchmarks compared to the baselines.

**Strengths:**

1.The paper visually demonstrates through cosine similarity plots and Grad-CAM heatmaps that hallucinated image-text pairs tend to exhibit lower alignment and misdirected visual attention compared to non-hallucinated pairs.

2.The proposed method, involving the construction of negative image samples via AHAF and subsequent contrastive finetuning ACFT, is shown to improve the performance of the tested models LLaVA1.5 and MiniGPT-4 on the POPE and MME-Existence hallucination benchmarks.

**Weaknesses:**

1.The core ideas identifying misalignment between visual and text representations as a cause for hallucination , and employing contrastive learning to improve alignment and mitigate hallucination —may lack significant novelty, as similar concepts have been discussed in prior work HACL[1].

[1]Hallucination Augmented Contrastive Learning for Multimodal Large Language Model

2.A major concern lies in the choice of base models for experimentation. The paper evaluates its method on LLaVA1.5 and MiniGPT-4 , which are relatively old, approx. 2 years and significantly weaker compared to the current sota in VLMs. Given the rapid iteration cycles in VLM development, it remains unclear whether the proposed method would yield similar benefits for much more capable models like Qwen2.5VL or InternVL3. The lack of validation on contemporary models severely limits the practical relevance and generalizability of the findings.

**Questions:**

1.The evaluation of hallucination mitigation primarily relies on the POPE and MME-Existence benchmarks, which predominantly use binary or multiple-choice formats to assess object presence . Object hallucination, however, also manifests significantly in free-form text generation like image captioning. Could the authors provide results on caption-based hallucination benchmarks, such as CHAIR, to demonstrate the effectiveness of ACFT in reducing hallucinations in generated descriptive text, and quantify the performance improvement in that setting?

---

> ### Author Response · Authors · 2025-11-22
> **Official Comment by Authors (1/3)**
>
> **Weakness 1: Difference between HACL and ACFT**
>
> We thank the reviewer for raising the concern about novelty.
>
> While both HACL and our ACFT use contrastive learning to mitigate hallucination, they actually approach the hallucination problem from two different perspectives and are complementary to each other. **Their motivation and method design are fundamentally different**.
>
> ***Motivation:***
>
> HACL is motivated from a **text perspective**: it attributes hallucination to unsatisfactory cross-modal alignment and the entanglement between representations of hallucinative and non-hallucinative texts. Therefore, HACL designs a contrastive objective that **operates mainly in text space, with visual features kept fixed**.
>
> In contrast, ACFT is motivated by the observation that object hallucinations often arise from **incorrect extraction of image features and misalignment between image and text embeddings**, especially in short-answer settings where the model relies more heavily on the visual information. Thus, we analyse and mitigate hallucination from the **image-prior perspective**.
>
> ***Method Design:***
>
> The difference in motivation leads to differences in both dataset construction and training strategy. In HACL, authors generate **hallucination-augmented texts** to enlarge the gap between hallucinative and non-hallucinative text representations, while the visual input remains unchanged; its contrastive loss is thus designed to “push apart” hallucinative vs. non-hallucinative texts.
> In ACFT, we instead **fix the ground-truth text anchor** and **construct positive–negative image pairs** with AHAF. The contrastive loss is then defined between the anchor text and these paired images, and is tailored to make the model sensitive to subtle visual differences.
>
> Finally, HACL is proposed as a **pre-training method**. It relies on large-scale data and a huge amount of computation resources (100% of the dataset). While ACFT is a **lightweight fine-tuning method**: by directly mining adversarial examples that expose the model’s weakest points, we achieve strong improvements with only about 6k samples (around 0.9% of the COCO dataset). This compute-efficient, image–driven contrastive fine-tuning regime is, to our knowledge, not covered by HACL.

---

> > ### Author Response · Authors · 2025-11-22
> > **Official Comment by Authors (2/3)**
> >
> > **Weakness 2: The base models for experimentation are relatively old. ACFT should be  validated on contemporary models**
> >
> > Thank you for pointing this out. We agree that our original choice of base models is somewhat outdated. We chose the two base models because they are widely used in prior hallucination studies and offer a clear comparison point to existing baseline methods. To directly address your concern, we have implemented ACFT on two recent LVLMs, **Qwen2.5-VL-7B** and **InternVL-3.5-4B**, and evaluated them on POPE and MME benchmarks.
> >
> > **Results for Qwen2.5VL 7B**
> > | **Benchmark** | **Subset**      | **Method** | **ACC** | **Precision** | **Recall** | **F1 Score** |
> > |---------------|-----------------|-----------|--------:|--------------:|----------:|------------:|
> > | **POPE**      | *Adversarial*   | original  | 0.864   | **0.940**     | 0.778     | 0.851       |
> > |               |                 | **ACFT**  | **0.877** | 0.897       | **0.852** | **0.874**   |
> > | **POPE**      | *Popular*       | original  | 0.875   | **0.965**     | 0.778     | 0.861       |
> > |               |                 | **ACFT**  | **0.900** | 0.942       | **0.852** | **0.895**   |
> > | **POPE**      | *Random*        | original  | 0.884   | **0.987**     | 0.778     | 0.870       |
> > |               |                 | **ACFT**  | **0.916** | 0.976       | **0.852** | **0.910**   |
> > | **MME**       | *Existence*     | original  | **1.000** | **1.000**   | **1.000** | **1.000**   |
> > |               |                 | **ACFT**  | **1.000** | **1.000**   | **1.000** | **1.000**   |
> > | **MME**       | *Whole*         | original  | 0.870   | 0.836         | **0.920** | **0.878**   |
> > |               |                 | **ACFT**  | **0.874** | **0.928**   | 0.810     | 0.865       |
> >
> > **Results for InternVL3.5 4B**
> > | **Benchmark** | **Subset**      | **Method** | **ACC** | **Precision** | **Recall** | **F1 Score** |
> > |---------------|-----------------|-----------|--------:|--------------:|----------:|------------:|
> > | **POPE**      | *Adversarial*   | original  | 0.863   | 0.835         | **0.905** | 0.869       |
> > |               |                 | **ACFT**  | **0.876** | **0.864**   | 0.893     | **0.878**   |
> > | **POPE**      | *Popular*       | original  | 0.899   | 0.894         | **0.905** | 0.899       |
> > |               |                 | **ACFT**  | **0.912** | **0.927**   | 0.895     | **0.911**   |
> > | **POPE**      | *Random*        | original  | 0.933   | **0.969**     | 0.845     | 0.930       |
> > |               |                 | **ACFT**  | **0.937** | 0.966       | **0.905** | **0.935**   |
> > | **MME**       | *Existence*     | original  | **1.000** | **1.000**   | **1.000** | **1.000**   |
> > |               |                 | **ACFT**  | **1.000** | **1.000**   | **1.000** | **1.000**   |
> > | **MME**       | *Whole*         | original  | 0.859   | **0.913**     | 0.795     | 0.850       |
> > |               |                 | **ACFT**  | **0.862** | 0.894       | **0.821** | **0.856**   |
> >
> > As shown in the Table, for both Qwen2.5-VL and InternVL 3.5, **ACFT delivers consistent gains on POPE and MME, even though the base models are already very strong**. (On MME Existence, the original models already achieve 100% accuracy. This reflects that the task is relatively easy for modern VLMs, leaving no headroom for further improvement.) **Overall, these results support our claim that ACFT is architecture-agnostic and transfers well to recent LVLMs.**
> >
> > We have included these results in our revised paper in Appendix A.11.

---

> > > ### Author Response · Authors · 2025-11-22
> > > **Official Comment by Authors (3/3)**
> > >
> > > **Question 1: Provide evaluation results on caption-based hallucination benchmarks**
> > >
> > > We thank the reviewer for the insightful comment.
> > >
> > > Firstly, we want to clarify **why we focused on binary benchmarks**. As discussed in the Introduction section, most existing works analyze object hallucination from the text side and attribute hallucination mainly to over-reliance on language priors, while the visual dimension remains relatively underexplored. This motivates us to study and mitigate object hallucination from the image-prior perspective, which is different from most previous works. Binary “Yes/No” benchmarks are particularly suitable for this purpose: their short answers reduce the influence of language priors. These benchmarks help us isolate hallucinations that arise from visual features or multimodal misalignment rather than from long-range text generation. Our work does not conflict with previous studies; instead, they approach the problem from two different perspectives and are actually complementary to each other.
> > >
> > > We fully agree with the reviewer that showing generalization to open-ended, caption-level hallucination is important. Thus, we have conducted experiments on five description-level benchmarks: **CHAIR, CCEval, AMBERA, MMHal-Bench, and ObjectHal**. For CHAIR, we sample 500 images from COCO 2014 val, prompt the model with “Please describe this image in detail.”, and set max_new_token to 512. For CCEval, AMBERA, MMHal-Bench, and ObjectHal, we follow each benchmark’s original evaluation setting.  All experiments are conducted on LLaVA v1.5-7B.
> > >
> > > | **Benchmark**            | **Metric**                    | **Original** | **ACFT**       |
> > > |--------------------------|-------------------------------|-------------:|---------------:|
> > > | **CHAIR**                | CHAIR$_s$ ${\downarrow}$      | 0.508        | **0.494**      |
> > > |                          | CHAIR$_i$ ${\downarrow}$      | 0.142        | **0.137**      |
> > > | **CCEval**               | CHAIR$_s$ ${\downarrow}$      | 0.870        | **0.850**      |
> > > |                          | CHAIR$_i$ ${\downarrow}$      | 0.348        | **0.328**      |
> > > | **AMBERA Generative**    | CHAIR ${\downarrow}$          | 0.112        | **0.089**      |
> > > |                          | Hal ${\downarrow}$            | 0.488        | **0.483**      |
> > > |                          | Cover ${\uparrow}$            | **0.518**    | 0.513          |
> > > |                          | Cog ${\downarrow}$            | 0.047        | **0.043**      |
> > > | **AMBERA Discriminative**| ACC ${\uparrow}$              | 0.716        | **0.729**      |
> > > |                          | Precision ${\uparrow}$        | 0.933        | **0.941**      |
> > > |                          | Recall ${\uparrow}$           | 0.617        | **0.630**      |
> > > |                          | F1 ${\uparrow}$               | 0.743        | **0.755**      |
> > > | **MMHal-Bench**          | Score ${\uparrow}$            | **2.710**    | 2.650          |
> > > |                          | Hallucination Rate ${\downarrow}$ | 0.604   | **0.594**      |
> > > | **ObjectHal**            | Response Hal ${\downarrow}$   | 0.568        | **0.562**      |
> > > |                          | Obj Hal ${\downarrow}$        | 0.283        | **0.277**      |
> > >
> > > The results show that, although ACFT is trained **only** on short-answer data, it consistently reduced caption-level hallucination **across all five benchmarks**, with lower CHAIR scores and hallucination rates compared to the original model. This supports our core claim that strengthening visual perception and multimodal alignment **benefits not only binary settings but also transfers to open-ended description tasks**. ACFT enables the model to focus more tightly on the target region, and form more discriminative representations for present versus absent objects in the embedding space. Thus, the model  is less likely to introduce nonexistent objects even when producing long, free-form captions.  Although the absolute gains on caption-level benchmarks are smaller than those on binary benchmarks, it is unsurprising given that long-form generation is still strongly affected by language priors. We believe these results show that improving visual grounding and multimodal alignment on short-answer tasks can provide a robust backbone that generalizes well to open-ended description generation.
> > >
> > > We have included these new caption-level results in the revised version in Appendix A.8.

---

### Official Review · Reviewer_yorZ · 2025-10-28

**Soundness:** 2
**Presentation:** 3
**Contribution:** 2
**Rating:** 4
**Confidence:** 3

**Summary:**

This paper introduces Adversarial Contrastive Fine-Tuning (ACFT) to mitigate object hallucination in Large Vision-Language Models (LVLMs). The authors first analyze the internal causes of hallucination via cosine similarity and Smooth Grad-CAM, observing that hallucinated samples exhibit weaker image–text alignment and diffuse visual attention. Based on these insights, they propose a two-stage framework:
(1) Adversarial Hallucination Attribute Flipping (AHAF) — a PGD-based adversarial procedure that generates positive/negative image pairs differing in final output generation.
(2) ACFT, which performs contrastive fine-tuning to maximize the similarity between text anchors and positive images while minimizing it for adversarial negatives.
Experiments on POPE and MME benchmarks using LLaVA-v1.5-7B and MiniGPT-4-13B show improved F1 and accuracy over baselines such as VCD, OPERA, VTI, and Woodpecker, without degrading general visual comprehension.

**Strengths:**

The paper’s main contribution is the idea of adversarially aligned positive-negative pairs for contrastive LVLM fine-tuning—uniting adversarial robustness and cross-modal alignment.

The methodology is detailed and mathematically rigorous, defining clear objectives for adversarial perturbation (Eq. 1–3) and the contrastive loss (Eq. 4–8).

Experiments are well-presented, including multiple baselines, two architectures, and ablations (with/without contrast loss, Grad-Cam).

The authors also show that the method uses only 0.9% of COCO for fine-tuning—a good efficiency advantage.

The paper is organized and easy to follow.

**Weaknesses:**

The central motivation—that low cosine similarity between image and text embeddings correlates with hallucination—remains empirically stated rather than conceptually justified. It is unclear why a higher similarity between the image feature and the corresponding text feature should necessarily indicate reduced hallucination. For instance, when the input question is semantically irrelevant to the image, one would still expect low similarity without this implying hallucination.

The evaluation is limited to binary yes/no benchmarks (POPE and MME-Existence), which mainly assess short-text object *existence detection*. However, since the core problem concerns object hallucination—i.e., generating descriptions of nonexistent objects in images—additional description-level (or caption-level) benchmarks (e.g., CHAIR, CCEval) are necessary for a more comprehensive evaluation.

The baselines are largely non-fine-tuned or inference-time methods, while ACFT involves model fine-tuning. This makes the comparison potentially unfair. The paper should include results against fine-tuned or post-training alignment baselines to ensure a fair assessment of relative improvement.

The Grad-CAM attention maps presented in the ablation/visualization (Figure 5) do not show statistically significant or consistently noticeable improvements, as the paper claimed " tightly focused on the target objects."

Although the paper claims efficiency advantages (using only 0.9% of COCO), it provides no quantitative metrics such as training time, GPU hours, or per-batch cost. Since each batch involves iterative adversarial optimization (PGD steps), the computational overhead could be substantial. A cost–performance analysis is necessary to validate the claimed efficiency benefits.

**Questions:**

How does the similarity metric behave when the question or text prompt is semantically irrelevant to the image? Do LVLMs consistently hallucinate under such conditions, or does low similarity simply reflect unrelated semantics?

Since object hallucination often manifests at the caption or descriptive level, could the authors extend evaluation to captioning-based hallucination benchmarks (e.g., those using CHAIR or CCEval) to test the generality of ACFT beyond binary yes/no tasks?

Given that ACFT involves model fine-tuning, can the authors include comparisons against other fine-tuned or post-training alignment methods?

What is the runtime overhead of the framework?

---

> ### Author Response · Authors · 2025-11-22
> **Official Comment by Authors (1/4)**
>
> **Weakness 1 && Question1: The link between low image–text cosine similarity and hallucination is only empirically observed, not conceptually justified, and may not hold when the question is irrelevant to the image.**
>
> We appreciate the reviewer’s feedback.
>
> Firstly, we would like to clarify what we mean by “ground-truth text feature” in our cosine-similarity analysis. In the analysis, we do not measure similarity between the image and an arbitrary input question. Instead, for each image, we first construct 10 POPE-style questions（ “Is there a [object] in the image?” ）based on the ground-truth annotations in COCO, and prompt the model to answer them. If all questions are answered correctly, we label the image as non-hallucinated; otherwise, we label it as hallucinated. Next, we randomly sample 500 hallucinated images and 500 non-hallucinated images. For each sampled image, we take its **ground-truth caption** and compute the cosine similarity between the image embedding and ground-truth text embedding. In hallucinated cases, the visual representation tends to be less consistent with the ground-truth textual description, leading to weaker alignment in the joint embedding space. This yields a clear gap in average cosine similarity between non-hallucinated and hallucinated samples (0.158 vs. −0.122). Under this setup, the reviewer’s example of a semantically irrelevant question does not arise: **our analysis is explicitly conditioned on text that is intended to correctly describe the image**, and it is conducted on datasets (COCO) where image–text relevance is enforced by construction. We also want to stress that we do not claim cosine similarity to be a universally reliable metric for hallucination, and we agree that using it as an evaluation metric would indeed face the limitation pointed out by the reviewer. In our work, cosine similarity is used only within a specific experimental setup. It only functions as an analysis tool to reduce image-prior hallucinations under controlled conditions, not as a comprehensive metric for all forms of hallucination in LVLMs.
>
> Secondly,  we want to stress that the cosine-similarity analysis is not purely empirical. This is consistent with how LVLMs are trained: their visual and textual encoders are typically pre-trained with CLIP-style contrastive objectives, which pull matched image–text pairs closer and push mismatched pairs apart. When the model answers correctly, the semantics of the text (e.g., “there is a truck”) are supported by the visual evidence and thus form a matched pair; when the model hallucinates an object that is absent from the image, the resulting image–text pair is effectively mismatched, which naturally leads to reduced similarity. This provides a **model structural reason** why we expect a statistical correlation between cosine similarity and hallucination in such models.
>
> We have included a detailed description for the cosine similarity analysis in our revised manuscript in Appendix A.1.1.

---

> ### Author Response · Authors · 2025-11-22
> **Official Comment by Authors (2/4)**
>
> **Weakness 2 && Question2: Lack of evaluation on description-level benchmarks**
>
> We thank the reviewer for the insightful comment.
>
> Firstly, we want to clarify **why we focused on binary benchmarks**. As discussed in the Introduction section, most existing works analyze object hallucination from the text side and attribute hallucination mainly to over-reliance on language priors, while the visual dimension remains relatively underexplored. This motivates us to study and mitigate object hallucination from the image-prior perspective, which is different from most previous works. Binary “Yes/No” benchmarks are particularly suitable for this purpose: their short answers reduce the influence of language priors. These benchmarks help us isolate hallucinations that arise from visual features or multimodal misalignment rather than from long-range text generation. Our work does not conflict with previous studies; instead, they approach the problem from two different perspectives and are actually complementary to each other.
>
> We fully agree with the reviewer that showing generalization to open-ended, caption-level hallucination is important. Thus, we have conducted experiments on five description-level benchmarks: **CHAIR, CCEval, AMBERA, MMHal-Bench, and ObjectHal**. For CHAIR, we sample 500 images from COCO 2014 val, prompt the model with “Please describe this image in detail.”, and set max_new_token to 512. For CCEval, AMBERA, MMHal-Bench, and ObjectHal, we follow each benchmark’s original evaluation setting.  All experiments are conducted on LLaVA v1.5-7B.
>
> | **Benchmark**            | **Metric**                    | **Original** | **ACFT**       |
> |--------------------------|-------------------------------|-------------:|---------------:|
> | **CHAIR**                | CHAIR$_s$ ${\downarrow}$      | 0.508        | **0.494**      |
> |                          | CHAIR$_i$ ${\downarrow}$      | 0.142        | **0.137**      |
> | **CCEval**               | CHAIR$_s$ ${\downarrow}$      | 0.870        | **0.850**      |
> |                          | CHAIR$_i$ ${\downarrow}$      | 0.348        | **0.328**      |
> | **AMBERA Generative**    | CHAIR ${\downarrow}$          | 0.112        | **0.089**      |
> |                          | Hal ${\downarrow}$            | 0.488        | **0.483**      |
> |                          | Cover ${\uparrow}$            | **0.518**    | 0.513          |
> |                          | Cog ${\downarrow}$            | 0.047        | **0.043**      |
> | **AMBERA Discriminative**| ACC ${\uparrow}$              | 0.716        | **0.729**      |
> |                          | Precision ${\uparrow}$        | 0.933        | **0.941**      |
> |                          | Recall ${\uparrow}$           | 0.617        | **0.630**      |
> |                          | F1 ${\uparrow}$               | 0.743        | **0.755**      |
> | **MMHal-Bench**          | Score ${\uparrow}$            | **2.710**    | 2.650          |
> |                          | Hallucination Rate ${\downarrow}$ | 0.604   | **0.594**      |
> | **ObjectHal**            | Response Hal ${\downarrow}$   | 0.568        | **0.562**      |
> |                          | Obj Hal ${\downarrow}$        | 0.283        | **0.277**      |
>
> The results show that, although ACFT is trained **only** on short-answer data, it consistently reduced caption-level hallucination **across all five benchmarks**, with lower CHAIR scores and hallucination rates compared to the original model. This supports our core claim that strengthening visual perception and multimodal alignment **benefits not only binary settings but also transfers to open-ended description tasks**. ACFT enables the model to focus more tightly on the target region, and form more discriminative representations for present versus absent objects in the embedding space. Thus, the model  is less likely to introduce nonexistent objects even when producing long, free-form captions. Although the absolute gains on caption-level benchmarks are smaller than those on binary benchmarks, it is unsurprising given that long-form generation is still strongly affected by language priors. We believe these results show that improving visual grounding and multimodal alignment on short-answer tasks can provide a robust backbone that generalizes well to open-ended description generation.
>
> We have included these new caption-level results in the revised paper in Appendix A.8.

---

> ### Author Response · Authors · 2025-11-22
> **Official Comment by Authors (3/4)**
>
> **Weakness 3 && Question3: Lack of comparison with post-training baselines**
>
> Thank you for raising this important concern.
>
> To address this, we have added 4 post-training baselines: (1) a **SFT baseline** that trains the same backbone as ACFT on the same data using a standard cross-entropy loss; (2) **LLaVA-RLHF**, we directly use the model weights released by the authors; (3) **OPA-DPO**, a DPO-based method for hallucination mitigation, we use the model weights released by the authors; and (4) **CHiP-DPO**, we reproduce the method using the authors’ released data and training scripts. We evaluate all of these methods under the same base model(LlaVA v1.5 7B) and report their performance on POPE, MME-Existence, and the full MME benchmark. The results are shown below.
>
> | **Benchmark** | **Subset**        | **Method**    | **ACC** | **Precision** | **Recall** | **F1 Score** |
> |---------------|-------------------|---------------|--------:|-------------:|----------:|------------:|
> | **POPE**      | *Adversarial*     | SFT           | 0.797   | 0.743        | **0.906** | 0.817       |
> |               |                   | LLaVA-RLHF    | 0.813   | 0.835        | 0.780     | 0.806       |
> |               |                   | CHiP-DPO      | 0.839   | 0.923        | 0.739     | 0.821       |
> |               |                   | opadpo        | 0.827   | **0.944**    | 0.697     | 0.801       |
> |               |                   | **ACFT**      | **0.841** | 0.802      | 0.905     | **0.850**   |
> | **POPE**      | *Popular*         | SFT           | 0.862   | 0.832        | **0.906** | 0.867       |
> |               |                   | LLaVA-RLHF    | 0.847   | 0.901        | 0.780     | 0.836       |
> |               |                   | CHiP-DPO      | 0.855   | 0.962        | 0.738     | 0.835       |
> |               |                   | opadpo        | 0.840   | **0.975**    | 0.698     | 0.813       |
> |               |                   | **ACFT**      | **0.906** | 0.907      | 0.905     | **0.906**   |
> | **POPE**      | *Random*          | SFT           | 0.896   | 0.888        | **0.906** | **0.897**   |
> |               |                   | LLaVA-RLHF    | 0.867   | 0.943        | 0.780     | 0.854       |
> |               |                   | CHiP-DPO      | 0.863   | 0.984        | 0.739     | 0.844       |
> |               |                   | opadpo        | 0.845   | **0.991**    | 0.696     | 0.818       |
> |               |                   | **ACFT**      | **0.897** | 0.890      | 0.905     | **0.897**   |
> | **MME**       | *Existence*       | SFT           | 0.950   | 0.935        | **0.967** | 0.951       |
> |               |                   | LLaVA-RLHF    | 0.967   | 0.967        | **0.967** | 0.967       |
> |               |                   | CHiP-DPO      | 0.967   | **1.000**    | 0.933     | 0.965       |
> |               |                   | opadpo        | 0.967   | **1.000**    | 0.933     | 0.965       |
> |               |                   | **ACFT**      | **0.983** | **1.000**  | **0.967** | **0.983**   |
> | **MME**       | *Whole*           | SFT           | 0.736   | 0.718        | 0.778     | 0.747       |
> |               |                   | LLaVA-RLHF    | 0.717   | 0.800        | 0.578     | 0.671       |
> |               |                   | CHiP-DPO      | 0.653   | **0.925**    | 0.334     | 0.491       |
> |               |                   | opadpo        | **0.753** | 0.897      | 0.572     | 0.699       |
> |               |                   | **ACFT**      | 0.747   | 0.683        | **0.922** | **0.785**   |
>
> The results show that, under a comparable data budget (approximately 6k samples), ACFT still outperforms these post-training baselines. This indicates that **the gains of ACFT do not simply come from “doing more fine-tuning,” but from the design of adversarially constructed positive–negative image pairs and the adversarial contrastive loss**, which more directly targets the visual misalignment underlying object hallucinations.
>
> We have added these post-training baselines and their results to our revised paper in Appendix A.9.

---

> > ### Author Response · Authors · 2025-11-22
> > **Official Comment by Authors (4/4)**
> >
> > **Weakness 4: The Grad-CAM visualizations do not clearly show “tightly focused” attention on target objects or show statistically convincing improvements.**
> >
> > Thank you for this comment. Our goal with the Grad-CAM visualization is to show that the attention distribution becomes more aligned with the presence or absence of the queried object, thereby reducing hallucinations.
> >
> > More concretely, we expect a semantically appropriate attention distribution to behave as follows:
> > - If the question asks about an object that exists in the image, the attention **should be focused on the target region**.
> > - If the question asks about an object that is absent, the attention **should be more dispersed**, rather than spuriously concentrating on an unrelated region (which tends to trigger hallucinations).
> >
> > Figure 5 illustrates these patterns, as desired. In **Cases 1 and 4**, the question refers to objects that are present. Before ACFT, the Grad-CAM maps are relatively dispersed and often miss the true target area; after ACFT, the attention becomes clearly more concentrated on the correct region, consistent with the corrected, non-hallucinated prediction. In contrast, in **Cases 2 and 3**, the question refers to objects that are absent. Before ACFT, the model’s attention concentrates on a wrong local region and the model hallucinates the queried object there; after ACFT, the attention becomes much more distributed, and the model correctly answers that the object is not present.
> >
> > We agree that purely qualitative inspection may not be fully convincing, so we additionally performed a quantitative analysis on the Grad-CAM maps. For each heatmap, we compute its **entropy**. We compute the entropy by building a histogram over attention heatmap intensity values. This histogram is treated as a probability distribution to calculate the Shannon entropy, which is further normalized by the maximum possible entropy (log of the number of bins). **Lower entropy indicates a more concentrated attention pattern, and higher entropy indicates a more distributed one.** The results are shown below.
> >
> > | Case | Before ACFT | After ACFT |
> > |------|-------------|------------|
> > | 1    | 0.836       | 0.763      |
> > | 2    | 0.721       | 0.774      |
> > | 3    | 0.782       | 0.832      |
> > | 4    | 0.803       | 0.782      |
> >
> >
> > The results are consistent with the above interpretation:
> > - In Cases 1 and 4 (object present), the entropy decreases after ACFT, indicating that the model’s attention becomes more focused on the true object region.
> > - In Cases 2 and 3 (object absent), the entropy increases after ACFT, indicating that the attention becomes more dispersed instead of over-confidently locking onto an incorrect region.
> >
> >
> > We have provided a more detailed explanation and analysis of Grad-CAM  in our revised paper in Appendix A.6.
> >
> > **Weakness 5 && Question 4:  Lack of a cost–performance analysis**
> >
> > Here we provide detailed analysis for both the data-construction and fine-tuning stages.
> >
> > ***Data construction cost ( AHAF stage)***:
> >
> > We construct a training set with 3,000 contrastive image samples and 3,000 normal samples. For the 3,000 contrastive samples, we run PGD on each image with 100 iterations. On **a single NVIDIA A100**, the average optimization time per image is about 15s, so constructing all 3,000 adversarial images takes **roughly 12 GPU-hours**. We want to clarify that PGD optimization is carried out only once in the offline data construction stage and does not affect training time cost. Our AHAF pipeline is fully automatic and directly mines adversarial examples that expose the model’s weakest points. In contrast, many post-training methods require manually collecting human preference data or calling external APIs for data augmentation.
> >
> > ***Training cost (ACFT stage)***:
> >
> > During fine-tuning, ACFT introduces **only a lightweight additional computation overhead** on top of standard SFT: we reuse the last-layer image and text representations to compute the adversarial contrastive loss, without extra forward passes through the backbone. In practice, for LLaVA-v1.5-7B with batch size 16 on a single A100, plain SFT training on our 6k-sample set takes **30 min 57s**, while ACFT takes **32 min 21s**. ACFT only introduced an overhead of about 90s, which we consider negligible. For comparison, when we run CHiP-DPO using the official training script on the same backbone, training requires 3 h 21 min 45 s on 4 A100 GPUs, i.e., more than **13 GPU-hours**, on a dataset of comparable scale.
> >
> > We have added the computational cost analysis to our revised paper in Appendix A.10.

---

### Author Response · Authors · 2025-11-22
**General Response**

To all reviewers:

We thank all the reviewers for their insightful comments and valuable suggestions! We are happy that the reviewers think that this work is “excellent” (TbAL), “technically sound” (xjhn, TbAL), and “demonstrates strong originality by creatively combining adversarial training with contrastive learning in a novel framework” (TbAL). The motivation is “well-articulated” (iJhQ) and “defines clear objectives”(yorZ). The method is “novel” (TbAL), “detailed and mathematically rigorous” (yorZ). The experiments are “well-presented” (yorZ) and the performance is “state-of-the-art” (TbAL), “improved” (xjhn), and “effective” (iJhQ), shows “high efficiency advantage without compromising general visual understanding” (yorZ, TbAL), making it “practical for real-world applications” (TbAL). The writing is “organized” (yorZ), “clear and coherent” (iJhQ), and “easy to follow” (yorZ, iJhQ). We address the comments below and have updated the paper accordingly.

Following the reviewers’ suggestions, we have made the following revisions to the paper:
- To address the reproducibility concern, we re-ran all baseline experiments and selected the optimal results for each baseline. We have updated the results in Tables 1 and 2 accordingly.
- We provide a detailed description of the cosine similarity analysis in Appendix A.1.1.
- We include a more detailed explanation and analysis for Grad-CAM in Appendix A.6
- We report evaluation results for description-level benchmarks in Appendix A.8
- We add comparison with post-training baselines in Appendix A.9
- We provide a computational cost analysis in Appendix A.10
- We include experiments on newer base models in Appendix A.11

All modifications have been highlighted in blue in the revised manuscript.

---

### Comment · Area_Chair_JuPh · 2025-11-26
**Author-Reviewer-AC Discussion (DDL: 12/3 9PM UTC)**

Dear Reviewers,

Thank you once again for your service to ICLR 2026. Now that the authors have submitted their rebuttal, I kindly ask you to take the following steps (if you have not done so already):

- Read the authors’ response and other reviews.
- Consider whether the rebuttal and additional comments affect your assessment of the paper. You may post the feedback to authors so that they can further follow up.
- Engage in interactive discussion with the authors -- **Note the Author-Reviewer-AC discussion period ends on 12/3 9PM UTC**. If you have more concerns/questions (e.g., requesting clarifications, new results), it is recommended to post your request asap, so that the authors have enough time to address them.

The current reviews for this paper are **mixed (scores: 4/4/8/4)**. Your further contributions are essential for forming a well-informed final decision.

I am happy to join and support the discussions between you and the authors. Please feel free to share your thoughts and participate actively in the discussion. Thanks!

Best regards,

AC

---

### Author Response · Authors · 2025-11-30

Dear AC:


To help you quickly grasp our rebuttal, we provide a brief summary below.


## Strengths of our paper as highlighted by the reviewers
We are happy that the reviewers think that this work is “excellent” (TbAL), “technically sound” (xjhn, TbAL), and “demonstrates strong originality by creatively combining adversarial training with contrastive learning in a novel framework” (TbAL). The motivation is “well-articulated” (iJhQ) and “defines clear objectives”(yorZ). The method is “novel” (TbAL), “detailed and mathematically rigorous” (yorZ). The experiments are “well-presented” (yorZ) and the performance is “state-of-the-art” (TbAL), “improved” (xjhn), and “effective” (iJhQ), shows “high efficiency advantage without compromising general visual understanding” (yorZ, TbAL), making it “practical for real-world applications” (TbAL). The writing is “organized” (yorZ), “clear and coherent” (iJhQ), and “easy to follow” (yorZ, iJhQ). During the discussion phase, reviewer TbAL further remarked that our work presents “at the very least, a clear contribution,” highlighted our image-prior analytical perspective as “notably novel compared to work in the same period on hallucination,” and found the results “convincing”. Reviewer iJhQ also emphasized that ACFT is “a sound and efficient method”.
## Revisions to the original paper
Following the reviewers’ suggestions, we have made the following revisions to the paper:
- To address the reproducibility concern, we re-ran all baseline experiments and selected the optimal results for each baseline. We have updated the results in Tables 1 and 2 accordingly.
- We provide a detailed description of the cosine similarity analysis in Appendix A.1.1.
- We include a more detailed explanation and analysis for Grad-CAM in Appendix A.6
- We report evaluation results for description-level benchmarks in Appendix A.8
- We add comparison with post-training baselines in Appendix A.9
- We provide a computational cost analysis in Appendix A.10
- We include experiments on newer base models in Appendix A.11


All modifications have been highlighted in blue in the revised paper.


## Reviewer concerns addressed during the rebuttal
In the rebuttal, we carefully addressed **all the concerns** raised by the reviewers. We (1) clarified the cosine-similarity analysis, (2) added five caption-level hallucination benchmarks, (3) compared ACFT against four post-training baselines, (4) strengthened our Grad-CAM analysis with quantitative Shannon-entropy metrics, (5) provided a detailed computational cost analysis for ACFT, (6) clarified the novelty of ACFT relative to HACL, (7) proved that ACFT continues to yield gains on stronger, more recent VLMs, (8) added cross-dataset evaluation and robustness experiments for ACFT, (9) resolved reproducibility concerns on baselines, (10) clarified and empirically validated our image-domain contrastive design, and (11) showed that combining ACFT with a language-prior method yields significant improvements on both discriminative and generative benchmarks, directly addressing concerns about limited gains on open-ended tasks.


All the concerns and where they are addressed are listed below:


- (1) Concern about cosine-similarity analysis (Reviewer yorZ) → Location: Response to Reviewer yorZ, Comment 1


- (2) Lack of caption-level evaluation (Reviewers yorZ, xjhn, TbAL, iJhQ)  → Location: Response to Reviewer yorZ, Comment 2


- (3) Lack of post-training baselines (Reviewers yorZ, iJhQ)  → Location: Response to Reviewer yorZ, Comment 3


- (4) Concern about Grad-CAM interpretation and persuasiveness (Reviewer yorZ) → Location: Response to Reviewer yorZ, Comment 4


- (5) Lack of efficiency and computational cost analysis (Reviewers yorZ, iJhQ)  → Location: Response to Reviewer yorZ, Comment 4 and Reviewer iJhQ, Comment 3


- (6) Novelty relative to HACL (Reviewer xjhn)  → Location: Response to Reviewer xjhn, Comment 1


- (7) Use of relatively old VLMs as base models (Reviewer xjhn) → Location: Response to Reviewer xjhn, Comment 2


- (8) Lack of cross-dataset validation and robustness of ACFT (Reviewer TbAL)  → Location: Response to Reviewer TbAL, Comment 2


- (9) Baseline reproducibility concerns (Reviewer iJhQ)  → Location: Response to Reviewer iJhQ, Comment 3


- (10) Conceptual limitation in contrastive design (Reviewer iJhQ)  → Location: Response to Reviewer iJhQ, Comment 4


- (11) Limited gains on open-ended benchmarks and scope of contribution (Reviewer iJhQ) → Location: Discussion with Reviewer iJhQ


We hope this structured summary can help you quickly locate the relevant parts of our rebuttal and appreciate that the main concerns of all reviewers have been thoroughly addressed, while the core positive assessments of ACFT’s soundness, novelty, and practicality remain intact.

---

### Meta-Review · Area_Chair_ucKb · 2026-01-07

**Summary:**

The paper targets object hallucination in large vision-language models and proposes Adversarial Contrastive Fine-Tuning. The authors diagnose hallucination as stemming from weaker image–text alignment in terms of cosine similarity gaps and problematic visual attention patterns. They then introduce a two-stage pipeline with a PGD-style adversarial process with finetuning. Experiments are presented to justify the performance of the algorithm.

The reviewers generally agree the effectiveness of the proposed algorithm, but are concerned with the novelty of this algorithm compared with the existing works. Reviewers' concern also covers the running time comparison especially with the PGD process.

**Reviewer Concerns:**

The authors provided a comprehensive rebuttal which solidify the VCD reproduction, and more comparisons compared with other finetuning algorithms, more benchmarks. However, the remaining concerns from the reviewers are (partially) open.

1. Reviewer xjhn mentioned the novelty of the proposed algorithm with HACL. Although the authors addressed part of this by declaring tat the HACL is based on text but the proposed methods are based on image input, the underlying methodology is still quite similar and need to elaborate more.
2. Some reviewers mentioned to add more experiments for modern LVLMs and raised the concern about the marginal improvement of the proposed methods. In authors rebuttal, the performance improvement, especially in the modern models, are indeed marginal. This also applies to to the cross-dataset evaluation raised by Reviewer TbAL.

**Reviewer Scores:**

Given the authors response, the three reviewers holding negative ratings are unlikely to upvote their results. This is acknowledged in iJhQ. Therefore the reviewers score change, if there was, will not significantly affect the decision of this paper.

---

### Decision · Program_Chairs · 2026-01-26

Reject